# Towards a task to assess boredom-like states in pigs–Stimulus validation as a basis

**Sara Hintze** *, **Heidi Heigl, Christoph Winckler**

Department of Sustainable Agricultural Systems, Institute of Livestock Sciences, University of Natural Resources and Life Sciences Vienna, Vienna, Austria

* sara.hintze@boku.ac.at

## Abstract

Animal boredom is a potentially prevalent, but underresearched animal welfare concern. To study the characteristics of boredom and its welfare consequences, we need to be sure that animals are actually bored and do not suffer from other negatively valenced states like apathy and depression. Animals' responses towards stimuli of different valence (positive, ambiguous, negative) have been suggested to help differentiating between these states. Apathetic animals are hypothesised to show a decreased interest in stimuli of all valences, whereas depressed animals are thought to be less interested in positive stimuli only, due to anhedonia, a key symptom of depression. In contrast, bored animals are hypothesised to show an increased interest in all types of stimuli, including negative ones. To ensure that the applied stimuli are indeed judged as positive, ambiguous or negative by the animals, we aimed to validate the valence of a range of stimuli in domesticated pigs, a species commonly kept under barren and monotonous conditions likely to induce boredom, as a basis for developing a task to distinguish between different negative states. Applying a within-subject design, 39 pigs (20 weaned piglets, 19 gilts) were individually exposed to twelve stimuli pre-classified as positive, ambiguous or negative in an approach-avoidance paradigm. The effects of stimulus, age (piglet, gilt) and their interaction on various approach and avoidance measures were analysed. Stimulus had a statistically significant effect on all measures and the observed pattern was according to pre-classification for many stimuli, but not all, resulting in a re-classification of the valence of five stimuli. The significant interaction between stimulus and age for some outcome measures indicates that age differences should be considered. Our study paves the ground for the selection of stimuli as well as outcome measures of future tasks aiming to differentiate between boredom, depression and apathy in pigs.

**Data Availability Statement:** Data are available from the Figshare database (https://doi.org/10.6084/m9.figshare.25512865.v1).

## 1 Introduction

Animal boredom has long been dismissed as a trivial concern, resulting in a lack of research and, consequently, little empirical evidence on if and how animals experience boredom-like states. However, the barrenness and monotony of the housing conditions of many of the

**Funding:** This research was funded in part by the Austrian Science Fund (FWF) [10.55776/T1164] with a grant awarded to SH. For open access purposes, the author has applied a CC BY public copyright license to any author accepted manuscript version arising from this submission. The funders had no role in study design, data collection and analysis, decision to publish, or preparation of the manuscript.

**Competing interests:** The authors have declared that no competing interests exist.

animals we keep on farms, in labs, in zoos or as pets, resemble conditions causing boredom in humans. Given the aversiveness of boredom in humans [1], animal boredom is thus a potentially very prevalent, yet understudied welfare concern [2,3]. Already in 1991 van Rooijen stated that "the danger of stress caused by boredom seems to be much more apparent under present husbandry conditions than stress caused by unpredictability" [4] and Wemelsfelder discussed boredom as a cause for the development of stereotypic animal behaviour [5]. Later, Duncan stressed the importance to better understand animal boredom, when proposing that "there is one state of suffering that requires further research and that is boredom [. . .]. Much remains to be done" [6]. However, only recently both conceptual e.g. [3,7] and empirical research e.g. [8–10] on animal boredom has gained momentum in animal welfare science.

Definitions of animal boredom have been derived from human psychology. For example, boredom in humans has been characterised by disengagement, high arousal, low arousal, inattention and altered time perception [11], five characteristics that may also play a role in animal boredom [7]. Moreover, boredom has been described as a lack of meaningful engagement [12]. Based on these definitions of human boredom and aiming to study boredom empirically in mink, Meagher and colleagues operationally defined boredom as a negative state caused by a barren environment that results in an increased interest in all kinds of stimuli, independent of their valence and thus independent of whether the stimuli are perceived as positive, ambiguous or negative [9,13]. Based on this definition, the researchers aimed to differentiate between boredom and other negative states associated with low activity, namely apathy and depression, that have both been associated with impoverished environments in animals [14,15]. Meagher and colleagues hypothesised that apathetic animals would show a decreased interest in stimuli of all valences as a result of reduced goal-directed behaviour, whereas depressed animals would be less interested in positive stimuli only, due to anhedonia, a key symptom of depression [16]. In contrast, bored animals were hypothesised to show an increased interest in all types of stimuli, including negative ones, which they would usually avoid [9]. This hypothesis was based on the notion that boredom results from being mentally unoccupied or unengaged, thus signalling the need to act to occupy our minds [12]. The need to become mentally engaged may lead to interactions with stimuli in the environment that would not be assessed as interesting or would even be perceived as negative under different circumstances. In two independent studies, Meagher and colleagues showed that mink housed in non-enriched cages showed an increased interest in all types of stimuli compared to mink housed in enriched cages, consistent with the authors' operational definition of boredom [9,13]. More recently, a third study in mink supported the findings of the first two studies showing that contact duration with the presented stimuli was reduced and orientation towards the stimuli tended to be reduced after mink had been moved to enriched housing, thereby also suggesting that boredom is a state that can be reduced by provision of enrichment [8]. Additionally, Burn and colleagues found that ferrets showed reduced contact time with negative and ambiguous stimuli when having access to a room with different enrichment items and a familiar human to interact with compared to control situations, indicating reduced signs of boredom in animals when receiving playtime [10].

Studying responses to stimuli of different valences is a promising approach for operationalising and empirically assessing boredom-like states in non-human animals [9]. However, before applying this approach to other species, validation of the valence of the used stimuli is crucial [9]. Whether stimuli are perceived as positive, ambiguous or negative has been empirically investigated based on approach and avoidance behaviour, e.g. in pigs [17] and sheep [18]. Whereas approach behaviour (e.g. orientation towards the stimulus, stimulus contact, etc.) is hypothesised to indicate positive perception of the stimulus, avoidance behaviour (e.g. escape attempts, signs of fear) is hypothesised to indicate negative perception of the stimulus. Some

researchers have suggested to further distinguish between promotion approach/avoidance and prevention approach/avoidance [19].

The aim of this study was to validate the valence of different stimuli (presumably positive, ambiguous, negative) in domesticated pigs (*Sus scrofa domestica*), a species commonly kept under barren and monotonous conditions, as a basis to develop a task that helps differentiating between different negatively valenced states. Specifically, we aimed to assess pigs' reactions to these stimuli without prior training or conditioning, since the spontaneous assessment of the stimuli is a precondition for future tests. Different to previous studies applying the approach-avoidance paradigm, we did not present presumed positive and negative studies simultaneously as e.g. in [18], nor did we test the aversion of stimuli while pigs consumed a food reward as e.g. in [17]. Instead, only one stimulus was presented at a time to a single pig without any other distraction. This procedure was chosen in order to validate the stimuli in a similar context as they would be later used to assess boredom and other negatively valenced states. In an experimental apparatus with three compartments (Start Room, Runway for stimulus presentation, Avoidance Room), 39 pigs were individually confronted with twelve stimuli each and various parameters were recorded regarding the animals' approach and avoidance behaviour. Since pigs' responses towards different stimuli may depend on their age and developmental stage, we tested weaned piglets and gilts.

## 2 Animals, material and methods

### 2.1 Animals and housing

All animals included in this study were born and raised at the pig facility Medau (VetFarm) of the University of Veterinary Medicine Vienna, where the study was conducted. Across two batches, 20 weaned piglets (10 females, 10 males) and 19 gilts were included. The origin of the piglets was either Large White x Piétrain or Large White x Large White whereas the origin of all gilts was Large White x Large White. Piglets were weaned with four weeks of age and selected from 20 different litters. Gilts from batch 1 were selected from two litters (three and five siblings) whereas gilts from batch 2 were selected from four litters (2 x two, 1 x three and 1 x four siblings). The average age at time of testing and the average body weight when pigs were enrolled in the experiment per age group and batch is given in Table 1. The ten weaned piglets per batch were divided into two groups of five piglets each (one group with three males and two females, the other group with three females and two males). Each group was housed in a 4.70 x 3.30 m partially slatted pen (3.1 m² per piglet) with a heated and covered lying area, a round trough and nipple drinkers. Piglets were fed *ad libitum* and received fresh sawdust on a daily basis. The eight gilts of batch 1 were divided into two groups of four and each group was housed in a 3.2 x 4.4 m partially slatted pen (3.5 m² per gilt). The eleven gilts of batch 2 were divided into two groups of 5 and 6 animals with each group housed in an 8.80 x 2.44 m partially slatted pen (4.3 m² per gilt for the group of 5 animals and 3.6 m² per gilt for the group of

**Table 1. Number and age at time of testing of piglets and gilts in both batches.**

| Batch | Piglets | | | Gilts | | |
|---|---|---|---|---|---|---|
| | Number | Age [weeks] | Weight [kg ± SD] | Number | Age [weeks] | Weight [kg ± SD] |
| 1 | 10 | 7–8 | 7.4 ± 1.6 | 8 | 16–17 | 56.9 ± 7.9 |
| 2 | 10 | 8–9 | 8.1 ± 1.3 | 9 | 13–14 | 35.9 ± 2.7 |
| | | | | 2 | 18–19 | 64.1 ± 4.9 |

Age is given for the start of testing whereas weight is given for the time when pigs were enrolled in the study (i.e. at the start of habituation). SD: Standard deviation.

6 animals). All pens were equipped with a longitudinal trough, nipple drinkers and two wooden logs fixed on the wall by a chain. Gilts were fed twice daily and had a minimal amount of chopped straw. We only included gilts in the older age group since no fattening pigs and thus no older male animals were kept in the pig facility during the time of the experiment.

## 2.2 Test apparatus

The experiment was conducted in a wooden test apparatus. Fig 1 shows a schematic overview of the side and top views of the apparatus. The apparatus consisted of a Start Room (1.5 x 1.5 m) with an entrance/exit gate and two guillotine doors (Figs 1 and 2A) and a Runway for presentation of the stimuli (4 x 1.5 m; Figs 1, 2C and 2D). It was placed in the aisle of an empty fattening unit so that the side guillotine door of the Start Room was adjacent to an existing partially slatted pen, which was used as Avoidance Room (4.4 x 2.4 m; Figs 1 and 2B). The

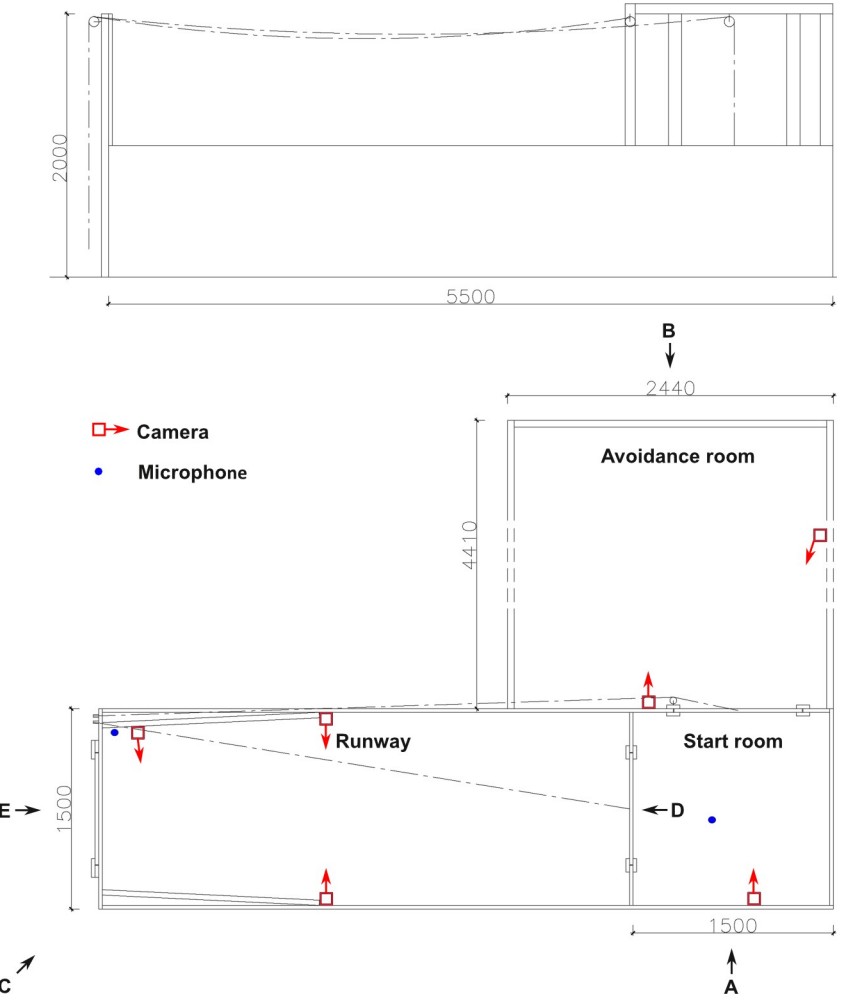

**Fig 1.** Schematic drawing of the test apparatus from the side (A) and top (B). Positions of the cameras (red square and arrow) and microphones (blue dots) are shown. Black arrows with capital letters indicate the camera perspectives from which the pictures presented in Fig 2A–2E were taken. The guillotine doors between the Start Room and the Runway (also shown in Fig 2A and 2D (closed) and 2G (opened)), between the Start Room and the Avoidance Room (also shown in Fig 2A and 2B (closed) and 2G (opened)) and at the end of the Runway where the stimuli were presented (also shown in Fig 2C and 2D (closed) and 2E (opened)) are indicated as [–]. Dashed lines indicate the ropes used to open the guillotine doors via a pulley system (also visible in Fig 2D).

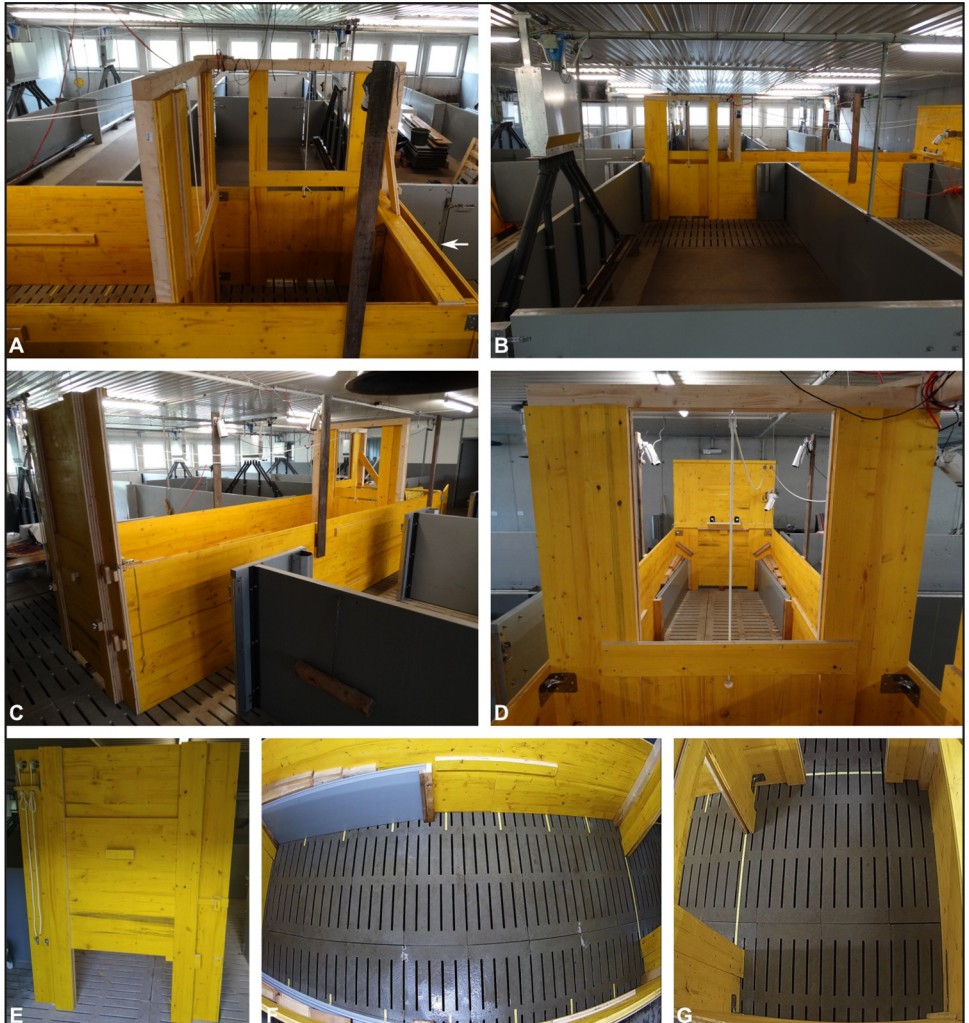

**Fig 2. Test apparatus.** A: Start Room with entrance/exit gate (on the right, see arrow) and guillotine doors to the Runway and the Avoidance Room. B: Avoidance Room. C: Runway from outside. D: Runway inside. E: Runway from outside. View on back wall with guillotine door and ropes for opening and closing the doors. F: Runway from the top with markings on the floor (every 50 cm). G: Start Room from top with markings to help indicating room changes. The perspectives of which pictures A – E were taken are presented in Fig 1.

guillotine doors between the Start Room and the Runway as well as the Start Room and the Avoidance Room could be operated with ropes and pulleys from the rear end of the Runway. Another guillotine door was located at the rear end to enable the experimenter to place the stimuli in the Runway (Figs 1 and 2E). On the inside of the rear end of the apparatus, above the guillotine door, a small shelf was installed for the permanently mounted speakers used to play test tones (Fig 2D, 2.4).

Using the NUUO® monitoring programme (NUUO-NVR-8D, 8-channel network recorder), six cameras, including two sound cameras with microphone (Santec SNC-441RBIAe) and four cameras without sound recording (2 x Dahua IPC-HFW2200RP-Z, Dahua IPC-HFW2221RP-ZS-IRE6, Santec SNC-441RBIAe) were installed to record the behaviour of the animals in the test apparatus (Fig 1), which could also be followed live on a monitor behind the test apparatus. The locations of the microphones shown in Fig 1 allowed capturing sounds from the complete test area. To avoid blind spots in the area of the stimulus

presentation, this area was conically narrowed with plastic walls (Fig 2D). Yellow adhesive strips were placed inside the Runway at 50 cm intervals and in the transitions between the apparatus rooms to facilitate estimating the distance to the stimulus and identifying compartment changes when coding the videos (Fig 2F and 2G).

## 2.3 Experimental design

The experiment was run in two batches, each including a habituation phase and a test phase (= stimulus presentation). Habituation lasted for twelve days for batch 1 and 13 days for batch 2 with a maximum of two habituation sessions per animal and day (in groups or individually, for more details see 2.3.1 and Table 2). The test phase took place for seven consecutive days with one day of break per batch. During the test phase, each pig was exposed to twelve different stimuli (four stimuli per presumably positive, ambiguous, negative valence; see 2.4) for three minutes each. During habituation and test phase, the pigs could not see or hear humans behind the experimental apparatus. The barn light was permanently on and the ventilation was turned off. No other pigs were in the room.

**2.3.1 Habituation.** Prior to the test phase, pigs were gradually habituated to the path between their home pens and the test room, all three compartments of the test apparatus (i.e. Start Room, Runway, and Avoidance Room), and the sounds of opening and closing of the guillotine doors. For this purpose, chocolate raisins were distributed on the floor of the test apparatus. The Avoidance Room was opened only after one to three sessions. The guillotine door to the Avoidance Room was opened one minute after the guillotine door to the Runway was opened, as it was done in the test phase, too. Thus, the pigs were familiarised with the whole experimental procedure including the associated sounds of the guillotine doors.

The group size of the pigs to be habituated together was gradually reduced from groups of five, four, three and two to individual habituation, depending on pigs' behaviour in the test apparatus. Habituation was deemed successful if pigs did no longer show attempts to exit the arena, restlessness or squealing. Table 2 lists the mean number of group and individual habituation sessions in weaned piglets and gilts per batch. The duration animals spent in the apparatus during habituation was reduced from ten to five minutes in groups and to three minutes in individual habituation sessions.

**2.3.2 Stimulus presentation.** Each pig was tested individually and confronted with each stimulus once. A maximum of two stimuli per animal per day were presented. The order of the twelve stimuli per animal was pseudorandomised, which means that the sequence of stimuli per pig allowed a maximum of two stimuli of the same valence to be tested consecutively, with no second test on the same day following a negative stimulus. The pseudorandomised lists were then adapted in a way to ensure that the sequence of stimuli was balanced across pigs. To

**Table 2. Number of habituation sessions (mean ± standard deviation) of weaned piglets and gilts per batch (1, 2), habituation stage (different group sizes, individual), duration per habituation session and if the Avoidance Room was open or closed.** Variation in group size (e.g. 4–5) is due to the different number of gilts per pen.

| Batch | Habituation stage (number of pigs) | Duration [Min.] | Avoidance Room open | Piglets | Gilts |
|---|---|---|---|---|---|
| 1 | group (4–5) | 10 | no | 3 ± 0 | 1 ± 0 |
| | group (2 – 3) | 10 | yes | 1 ± 0 | 1 ± 0 |
| | group (2–3) | 5 | yes | 3 ± 0.6 | 1.8 ± 0.5 |
| | individual | 3 | yes | 6.3 ± 0.5 | 7.4 ± 0.5 |
| 2 | group (2–5) | 10 | no | 2 ± 0 | 1.8 ± 0.4 |
| | group (2 – 3) | 10 | yes | 1 ± 0 | 1 ± 0 |
| | group (2–3) | 5 | yes | 5.2 ± 0.4 | 2 ± 0 |
| | individual | 3 | yes | 8.0 ± 0.0 | 7.4 ± 0.7 |

prevent pigs from habituating to recurrent patterns in the procedure, care was taken to ensure that the sequence of pigs per test day and the sequence of stimuli between pigs were different. Pigs were tested between 6:00 and 23:00. The testing order of pigs per day was first randomised using a random number generator in Excel and then adapted to be counterbalanced across test days.

For each test session, a pig was brought individually into the Start Room (with closed guillotine doors) and the entrance/exit gate was closed. Each test session was three minutes long and started as soon as the guillotine door between the Start Room and the Runway was opened. The pig then had the opportunity to enter the Runway and to explore the stimulus or to remain in the Start Room to avoid approaching the stimulus. After one minute, the guillotine door to the Avoidance Room was opened, allowing the pig to move further away from the stimulus. At the end of the three-minute test period, the pig was returned to its group and the test apparatus was cleaned with water and prepared for the next test session.

## 2.4 Stimuli

Aiming to select presumably positive, ambiguous and negative stimuli, we first screened the literature for potential stimulus candidates e.g. [17,20,21]. The most important selection criterion was that pigs could spontaneously assess the valence of the stimuli without the need to prior learn the meaning of a stimulus, as it is for example the case when they first need to get used to the taste of apples or chocolate. Twelve stimuli were selected consisting of four presumably positive, four presumably ambiguous and four presumably negative stimuli (Fig 3A–3O). The audio recordings of the Contact Calls, Restriction Calls, and Farm Noises (see below for more details) were kindly provided by Sandra Düpjan (Research Institute for Farm Animal Biology, Dummerstof, Germany) and Lisette Leliveld (University of Milan, Italy). Playback of all audio recordings was performed via a laptop and the permanently mounted loudspeakers (Fig 3E, 3K and 3O). Playbacks of Contact Calls, Restriction Calls and Farm Noises were played at different volumes to present all sounds as if they were coming from pigs present in the compartment, but not visible to the tested pig.

**2.4.1 Presumed positive stimuli.** The four presumably positive stimuli (Fig 1A–1E) included Silage, a mix of grass and corn silage (2 kg, ratio of grass and corn silage 1:1), Peat, a mix of peat (MOORSOL® PigletFitMoor/SOLAN) and corn kernels (3 kg, ratio of peat to corn kernels 5:1), a Mirror, and the audio recordings of various Contact Calls. Both Silage and Peat were offered on a yellow wooden tablet (51 x 51 x 7 cm; Fig 3A and 3B). This tablet was placed in front of the back wall of the Runway using two screws attached to the bottom of the tablet, which prevented pigs from moving it. Between sessions, Silage and Peat were stored in a separate room inaccessible to the pigs to limit the odour in the test room as much as possible to the test time of these stimuli. Neither piglets nor gilts had prior experience with silage or peat. The Mirror (70 x 50 cm) was hung from a wire on the back wall of the Runway (Fig 3C). Since pigs were able to move the Mirror back and forth, it was fixed with an adjustable wooden structure after the first test sessions (Fig 3D). The audio recordings of the Contact Calls originated from 20 pigs during periods of short social isolation and were shown to have a communicative function [22]. We did not expect these calls to be positive *per se*, but hypothesised that pigs in our study, who were isolated and did not hear conspecifics during testing, may perceive the calls of another pig as positive. Even though this hypothesis is speculative, we still decided to list the Contact Calls as a presumably positive stimulus with its true valence being tested in our experiment. Contact Calls were divided among all pigs such that each recording was played at least once in weaned piglets and gilts across both batches. The original one-minute audio recordings were edited using Audacity® to create three-minute audio clips with

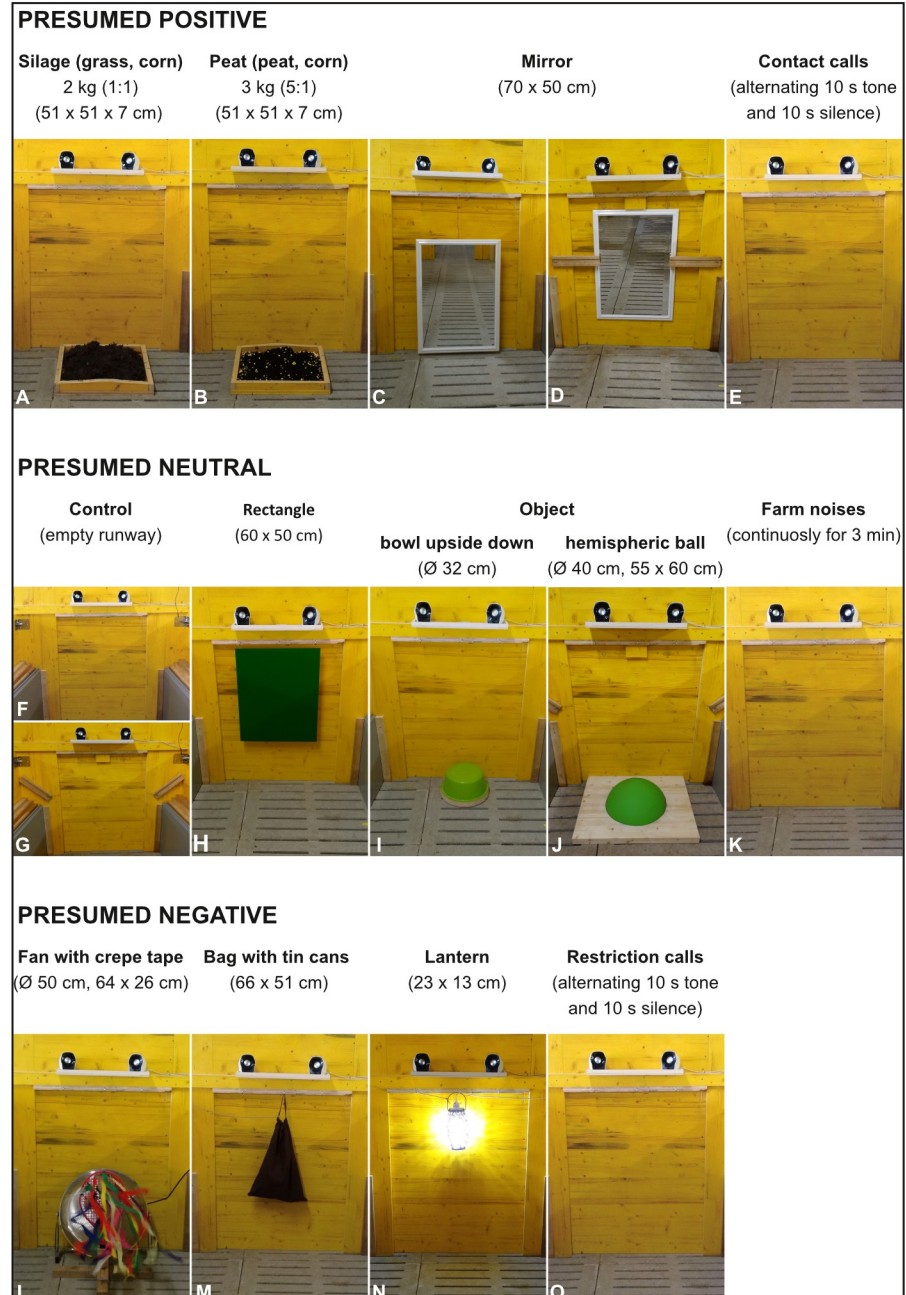

**Fig 3.** Overview of the presumed positive (A – E), ambiguous (F – K) and negative (L – O) stimuli presented in the Runway.

alternating ten seconds of audio and ten seconds of break. These breaks were inserted since it would be unnatural to have constant calling for three minutes without any breaks in-between; we thus aimed to make this stimulus resemble a more natural situation by inserting breaks. Any squealing was cut in the process to keep the recordings as comparable as possible.

**2.4.2 Presumed ambiguous stimuli.** The four stimuli defined as potentially ambiguous (Fig 3F–3K) encompassed a Control, i.e. the empty Runway, a green Rectangle, a green plastic Object, and an audio recording of Farm Noises. The green rectangle was a painted wooden

panel (60 x 50 cm) hung from the back wall of the Runway (Fig 3H). An inverted green bowl (ø 32 cm) mounted on wood was used as a plastic object in batch 1 (Fig 3). Like the yellow wooden tablet, the wood was fixed to the floor with two screws in the gaps between the slats, and it could be lifted with the trunk but slipped only slightly. In batch 2, a green semi-circular ball (ø 40 cm) mounted on a wooden plate (55 x 60 cm) and fixed to the floor with screws in the same way was used as the plastic object (Fig 3J). Replacement of the plastic bowl was necessary because it was destroyed several times by gilts from batch 1. The same audio recording of Farm Noises was played to all pigs for three minutes without pauses.

**2.4.3 Presumed negative stimuli.** The four presumed negative stimuli (Fig 3L and 3O) included a Fan with a rack containing crepe tape, a Bag filled with tin cans, a metal Lantern with a bright LED and audio recordings of Restriction Calls. In front of the Fan (ø 50 cm), a construction consisting of a tennis racket in a wooden stand (64 x 26 cm) and coloured crepe tapes (up to 80 cm long) attached to the tennis racket was placed (Fig 3L). At the start of the test, the fan was turned on and the crepe tapes fluttered throughout the test period. The Bag, made of brown fabric and filled with tin cans (66 x 51 cm), was hung from the Runway back wall and shaken back and forth from the back of the Runway back wall throughout the test duration using a pulley system (Fig 3M). The metal Lantern with built-in high luminosity LED light (23 x 13 cm) was also suspended and oscillated back and forth for three minutes via the same rope construction as the Bag (Fig 3N). The audio recordings of the Restriction Calls originated from 30 piglets individually restrained in a dog harness for another study [23]. They were divided by all pigs such that each recording was played at least once for piglets as well as for gilts across both batches. The original one-minute sound recordings were previously edited using Audacity® to create three-minute sound clips with ten seconds of sound and ten seconds of breaks in-between. As for the Contact Calls, these breaks were inserted since it would be unnatural to have constant calling for three minutes without any breaks in-between. All sounds from the three auditory stimuli were likely to be known by the pigs since they had probably encountered them earlier in their lives. Even though prior experience with these stimuli may have differed between pigs, grunts and squeals most likely had similar meaning to them due to the inherent significance of these calls for pigs.

## 2.5 Outcome measures

Outcome measures were derived from the literature and are described in Table 3. Not all approach and avoidance measures could be applied across all stimuli. For example, Stimulus Contact could not be assessed for the auditory stimuli, Control and Bag (for an overview see Table 4), but was still included since it is an important measure of approach. Aiming to compensate for the fact that not all measures could be applied across all stimuli, which renders direct comparison difficult, we applied a range of measures aiming to better understand the bigger picture. In addition to classical measures of approach and avoidance, we also included behavioural measures indicative of negative affective states (different vocalisations, elimination behaviour and startling) aiming to gain additional insight in what exposure to the different stimuli means to the pigs. We could not include behavioural measures indicative of positive affect due to a lack of validated positive indicators.

The additional behavioural measures indicative of negative affective state, i.e. vocalisations, elimination behaviour, and startle responses, were recorded live during the test sessions, during which the behaviour of the pigs could be followed on a monitor. All other parameters were collected from the video recordings using Mangold INTERACT®. Since stimuli were clearly visible on the video clips and some of the outcome measures also directly referred to the stimuli (e.g. Stimulus Contact), blinding for stimulus was not possible. Moreover, since the

**Table 3. Ethogram describing all outcome measures.**

| Outcome measure | Description | Recording | Interpretation* |
|---|---|---|---|
| Entering Start Room | All four claws have been placed over the transition mark between Runway and Start Room or Avoidance Room and Start Room and are now in the Start Room.<br> Entry into Start Room only possible after previous time spent in Runway or Avoidance Room (see below). | Frequency | Depends on context |
| Time spent in Start Room | Start: Guillotine door to the Runway is half-open (at the start of the test session) OR pig enters Start Room from another room (see above).<br> End: All four claws have been placed outside the Start Room, i.e. into the Runway or into the Avoidance Room. | Duration | Depends on context |
| Entering Runway | All four claws have been placed over the transition mark between the Start Room and the Runway and are now in the Runway. | Frequency | Approach |
| Time spent in Runway | Start: Pig enters Runway.<br> End: All four claws have been placed outside the Runway, i.e. into the Start Room. | Duration | Approach |
| Entering Avoidance Room | All four claws have been placed over the transition mark between the Start Room and Avoidance Room and are now in the Avoidance Room. | Frequency | Avoidance |
| Time spent in Avoidance Room | Start: Pig enters Avoidance Room.<br> End: All four claws have been placed outside the Avoidance Room, i.e. into the Start Room. | Duration | Avoidance |
| Escape Attempts | Attempt to leave the apparatus via the entrance/exit guillotine door of the Start Room and/or a specific corner in the Avoidance Room.<br>Duration:<br>Start:<br>*Entrance/exit guillotine door of the Start Room*: The pig's front legs are no more than 40 cm away from the entrance/exit door and the pig is facing towards the door.<br>*Corner in Avoidance Room*: Corner between apparatus wall and pen wall is defined by a 45 x 45 cm square. There is a narrow slit between the apparatus wall and the pen wall. The pig's front legs are inside the corner and it is facing the corner and/or touches corner walls with trunk and/or puts trunk through slit.<br>End: Front legs are no longer in the 40 cm range in front of the entrance/exit door or front legs are no longer in the 45 x 45 cm square of the corner of the Avoidance Room.<br>Frequency:<br>A new Escape Attempt is recorded if the behaviour stops for more than two seconds. | Duration Frequency | Avoidance |
| Stimulus Orientation | Start: Pig stands still in any of the apparatus rooms with the head facing the (source of the) stimulus. The trunk has no contact with the floor or apparatus (e.g. walls).<br>End: Pig starts touching the floor or apparatus walls with the trunk OR head is turned away from (the source of the) stimulus OR pig begins to approach stimulus OR pig turns away from stimulus.<br> Not recorded during Control. | Duration Frequency | Interest in/attention towards the stimulus |
| Stimulus Approach | Start: Pig moves towards the (source of the) stimulus; head is oriented towards the stimulus.<br>End: Pig stops moving towards the stimulus OR pig starts touching the floor or apparatus walls with the trunk OR head is turned away from stimulus OR pig turns away from stimulus.<br> Not recorded during Control, Contact Calls, Farm Noises, Restriction Calls. | Duration Frequency | Approach |

*(Continued)*

**Table 3.** (Continued)

| Outcome measure | Description | Recording | Interpretation* |
|---|---|---|---|
| Stimulus Contact | Physical contact with trunk (including sniffing, digging, eating) and/or front legs. Contact with other body parts is not recorded. Sniffing, digging and eating are recorded as Stimulus Contact even if the trunk is not in permanent contact with the stimulus (e.g. during chewing). Contact with material rooted out on the slatted floor (Silage, Peat) is also considered as contact.<br>Duration:<br>Start: Pig touches the stimulus with trunk and/or front legs.<br>End: Elapsed time since last contact longer than five seconds OR Turning away (see below).<br>Frequency:<br>A new Stimulus Contact is recorded if the behaviour stops for more than five seconds.<br>Not recorded during Control, Contact Calls, Farm Noise, Restriction Calls, Bag. | Duration<br>Frequency | Approach |
| Minimal Distance | Minimal distance of the tip of the front claws to the stimulus across the test session if no contact was made. For Mirror, Rectangle, Lantern and Fan: distance between claw tip and apparatus back wall. For Silage, Peat and Object: distance between the claw tip and the front side of the wooden tablet/frame.<br>Not recorded during Control, Contact Calls, Farm Noises, Restriction Calls, Bag. | Distance [cm] (estimated from video clips with the help of markings on the floor (every 50 cm). | Short distance: Approach,<br>Long distance: Avoidance |
| Turning Away | Change in body position (head-tail line) by at least 90° compared to the previous body position shown during Stimulus Orientation, Stimulus Approach or Stimulus Contact.<br>Not recorded during Control, Contact Calls, Farm Noise, Restriction Calls. | Frequency | Avoidance |
| Grunting | Dark, harsh, guttural (not tonal) sound. Can be short or long [24]. | Frequency | Depends on context |
| Squealing | Loud, forcefully emitted, high-pitched, shrill and tonal (brightly coloured) sound. Can be short or long. Seamless transitions from grunts to squeals are recorded as squeals and not as grunts [24]. | Frequency | Negative [25],<br>indicative of stress [26] |
| Barking | Loud, short impact sound, similar to a dog bark [24,27]. | Frequency | Alarm call: sudden disturbance, surprise [26] |
| Defecation/urination | A new defecation/urination event is recorded if the behaviour stops for more than 20 seconds [28]. | Frequency | Indicative of fear or stress [28] |
| Startling | Rapid, jerky movement followed by short or long immobility [24]. | Frequency | Often shown together with barking; sudden disturbance or surprise |

* Please note that the interpretation of some of the behaviours is not straight forward and depends on the context. Turning Away, for example, is interpreted as an avoidance behaviour but since it can only follow Stimulus Orientation, Stimulus Approach or Stimulus Contact, it can only be shown after the animal has shown some interest in the stimulus.

student conducting the experiment was also involved in designing the experiment (HH), she was not blind to the research questions and hypotheses. Prior to video analysis, the repeatability of video coding at different time points (intra-observer agreement) and between observers (inter-observer agreement) was tested (see 2.6). If not stated otherwise, outcome measures were recorded during all test sessions.

## 2.6 Testing for intra-observer and inter-observer agreement

Intra- and inter-observer agreement was tested for the 16 outcome measures that were recorded from video clips. To this end, 26 video clips were selected to test for agreement, one video clip per stimulus and batch. For the green plastic Object, one clip of the green plastic

**Table 4. Overview of the stimuli excluded in the analyses per outcome measure.**

| Outcome measure | Stimuli excluded in the analysis |
|---|---|
| Entering Start Room [#] | none |
| Time spent in Start Room [s] | none |
| Entering Runway [#] | none |
| Entering Avoidance Room [#] | none |
| Time spent in Avoidance Room [s] | none |
| Escape Attempts [#] | none |
| Stimulus Orientation [#] | Control |
| Stimulus Approach [#] | Contact Calls, Control, Farm Noises, Restriction Calls |
| Stimulus Contact [#] | Contact Calls, Control, Farm Noises, Bag, Restriction Calls |
| Stimulus Contact [s] | Contact Calls, Control, Farm Noises, Bag, Restriction Calls |
| Minimal Distance [cm] | Contact Calls, Control, Farm Noises, Bag, Restriction Calls |
| Grunting [#] | none |
| Barking [#] | none |
| Squealing [#] | none |
| Excretion [#] | none |

bowl and of the green semi-circular ball were analysed. To check for intra-observer agreement, videos were coded twice with nine weeks in-between the two codings.

## 2.7 Ethics statement

This study was approved by the Ethics and Animal Welfare Committee of the University of Veterinary Medicine, Vienna, Austria, in accordance with the University's guidelines for Good Scientific Practice (ETK-112/07/2020). We followed the ARRIVE 2.0 Guidelines [29]; see Supplementary Information for additional information with respect to the ARRIVE Essential 10.

## 2.8 Statistical analyses

All statistical analyses were run in R (RStudio version 2022.02.3, R version 4.2.1).

**2.8.1 Intra- and inter-observer agreement.** Intra- and inter-observer agreement were analysed with Intraclass Correlation Coefficients (ICC, function icc, package irr) to assess agreement between first and second scoring of HH (intra-observer agreement) and between HH and another assessor (inter-observer agreement). For both intra- and inter observer agreement, we used a two-way model with "single rater" as "type" and assessing absolute agreement [30].

**2.8.2 Collinearity between outcome measures.** All outcome measures were tested for association using a correlation matrix (function rcorr, package hmisc). In case of highly correlated outcome measures (R $\leq$ -0.7 or R $\geq$ 0.7), we decided to run further analyses with the outcome measures, which we expected to be more relevant as a measure of approach or avoidance. Time spent in the Runway was positively correlated with the duration of Stimulus Contact (r = 0.77) and negatively correlated with time spent in the Start Room (r = -0.72) and in the Avoidance Room (r = -0.80); it was thus excluded from further analyses. The frequency of Stimulus Approach and Turning Away were positively correlated (r = 0.80) and we selected Stimulus Approach for all further analyses. Frequency and duration of Escape Attempts (r = 0.76), Stimulus Orientation (r = 0.77) and Stimulus Approach (r = 0.79) were all positively correlated and we decided to run all further analyses with the frequency measures since they revealed better inter-observer agreement.

**2.8.3 Assessment of the effect of stimulus and age group on measures of approach and avoidance.**   Models were built based on the hypothesis that the different stimuli would differently affect the approach and avoidance measures, potentially interacting with age group. Thus, stimulus (factor with twelve levels) and age group (factor with two levels: piglet, gilt) as well as their two-way interaction were used as fixed effects. Not all outcome measures could be assessed for all stimuli (Table 4). To analyse the effect of stimulus on duration and frequency of Stimulus Contact, we excluded the stimuli Contact Calls, Control, Farm Noises, Bag and Restriction Calls. All auditory stimuli as well as Control were excluded since it is impossible to be in contact with these stimuli. A similar reasoning was applied for Bag even though we recorded attempts to get in contact. However, since such an attempt was only recorded once, we decided to exclude Bag from the analyses of Stimulus Contact. The same reasoning applied to the Minimal Distance where these five stimuli were also excluded from the analysis. For Stimulus Approach the auditory stimuli and Control were excluded. For Stimulus Orientation only Control was excluded since animals could orient towards all other stimuli, including the origin of the auditory cues.

Piglet sex was part of the randomised block design and was thus not analysed separately since we did not have a specific hypothesis for sex differences; however, for exploratory interpretation data are presented graphically (S1 and S2 Figs). For all fixed effects including the interactions, we used dummy variables with sum contrasts [31]. To obtain p-values we compared the full model to a model reduced by one main effect or the interaction. By comparing the full model to reduced models using sum contrasts, the p-value of the main effects can be interpreted even if the interaction is significant. To adequately reflect dependencies in the experimental design including repeated measures and nesting, random effects for all models were pig (1–39) nested in group/pen (1, 2) nested in age group (piglet, gilt) nested in batch (1, 2). Litter was not included as crossed random effect since piglets were all selected from different litters, which is why dependencies between the gilts with respect to litter could not be considered in the statistical models. Data of one gilt from batch 1 when exposed with the Object were not included in the analysis since this session was stopped after less than 1.5 minutes when the gilt destroyed the plastic bowl, resulting in overall 467 sessions.

Outcome measures recorded as durations were transformed into proportions of the three-minute session and analysed using Beta regressions (function: glmmTMB, package: glmmTMB). Outcome measures recorded as frequencies were analysed using generalised linear mixed-effects models (function: bglmer, package: blme, 'family': poisson). No models were run for Startling since this behaviour was only recorded in 18 sessions. For Squealing, Barking and Excretion we converted the frequency measures in binomial data (0: not recorded during a session, 1: recorded during a session) and ran binomial models (function: bglmer, package: blme, 'family': binomial) since these behaviours were too rarely recorded for a good fit of the count models. For the Minimal Distance we created a subset of the data including only sessions in which the pig had not been in contact with the stimulus. We aimed to run a linear mixed-effects model (function: blmer, package: blme) to assess how close pigs came to the stimulus, however, these models did not converge, probably due to the relatively small number of sessions in which pigs were not in contact with the stimulus (n = 126), which were not evenly distributed across stimuli. Alternatively, we ran the function lme (package: nlme), but without sum contrasts.

We defined the significance threshold as alpha = 0.05. However, to correct for multiple testing, we applied the Benjamini-Hochberg procedure to decrease the False Discovery Rate [32]. After correction, all p-values smaller than 0.004 were considered to be statistically significant. Data are graphically presented as boxplots using the function ggplot2 (package: ggplot2). Model estimates as well as their low and high confidence intervals were calculated with the function effect (package: effects) and are displayed in the boxplots.

# 3 Results

## 3.1 Intra- and inter-observer agreement

Intra-observer agreement was 'excellent' (ICC > 0.9; [28]) for all outcome measures obtained from video recordings except for the frequency of Stimulus Approach, for which the ICC was 'good' (ICC > 0.75 – 0.9; Table 5). The lower bounds of the 95% Confidence Intervals (CI) fell mostly into the category 'excellent' with the exception of both frequency and duration of Stimulus Approach and the frequency of Turning Away, for which the lower CI bounds were categorised as 'good'. Agreement between observers ranged from 'poor' (ICC < 0.5, two outcome measures) to 'moderate' (ICC > 0.5 – 0.75, six outcome measures), to 'good' (two outcome measures) and 'excellent' (six outcome measures). The two outcome measures with poor

**Table 5. Intra- and inter-observer agreement of all outcome measures, which were recorded on 26 video clips (excluding vocalisations, elimination behaviour and startle responses, which were recorded live), presented as intraclass correlation coefficients with 95% confidence intervals.**

| Outcome measure | Intra-/ inter-observer agreement | Intraclass Correlation Coefficient (ICC)* | 95% Confidence Interval |
|---|---|---|---|
| Entering Start Room [#] | Intra | 1.00 | 0.99 < ICC < 1.00 |
| | Inter | 1.00 | 0.99 < ICC < 1.00 |
| Time spent in Start Room [s] | Intra | 0.98 | 0.96 < ICC < 0.99 |
| | Inter | 0.67 | 0.39 < ICC < 0.84 |
| Entering Runway [#] | Intra | 1.00 | 1.00 < ICC < 1.00 |
| | Inter | 0.99 | 0.97 < ICC < 0.99 |
| Time spent in Runway [s] | Intra | 1.00 | 1.00 < ICC < 1.00 |
| | Inter | 0.63 | 0.32 < ICC < 0.82 |
| Entering Avoidance Room [#] | Intra | 1.00 | 1.00 < ICC < 1.00 |
| | Inter | 1.00 | 1.00 < ICC < 1.00 |
| Time spent in Avoidance Room [s] | Intra | 0.99 | 0.98 < ICC < 1.00 |
| | Inter | 0.70 | 0.44 < ICC < 0.86 |
| Escape Attempts [#] | Intra | 0.98 | 0.96 < ICC < 0.99 |
| | Inter | 0.67 | 0.28 < ICC < 0.85 |
| Escape Attempts [s] | Intra | 0.99 | 0.97 < ICC < 0.99 |
| | Inter | 0.42 | 0.06 < ICC < 0.69 |
| Stimulus Orientation [#] | Intra | 0.97 | 0.93 < ICC < 0.99 |
| | Inter | 0.94 | 0.88 < ICC < 0.97 |
| Stimulus Orientation [s] | Intra | 0.99 | 0.98 < ICC < 1.00 |
| | Inter | 0.55 | 0.22 < ICC < 0.77 |
| Stimulus Approach [#] | Intra | 0.83 | 0.66 < ICC < 0.92 |
| | Inter | 0.85 | 0.56 < ICC < 0.94 |
| Stimulus Approach [s] | Intra | 0.93 | 0.76 < ICC < 0.98 |
| | Inter | 0.48 | 0.11 < ICC < 0.73 |
| Stimulus Contact [#] | Intra | 0.98 | 0.94 < ICC < 0.99 |
| | Inter | 0.94 | 0.88 < ICC < 0.97 |
| Stimulus Contact [s] | Intra | 0.97 | 0.94 < ICC < 0.99 |
| | Inter | 0.64 | 0.35 < ICC < 0.82 |
| Minimal Distance [cm] | Intra | 1.00 | 1.00 < ICC < 1.00 |
| | Inter | 1.00 | 1.00 < ICC < 1.00 |
| Turning Away [#] | Intra | 0.92 | 0.83 < ICC < 0.96 |
| | Inter | 0.80 | 0.59 < ICC < 0.90 |

* All p-values were < 0.05.

agreement, the durations of Escape Attempts and Stimulus Approach, were not considered in further analyses; only the frequency measures of these outcome measures were used, for which the agreement was 'moderate' and 'good', respectively. The range of the 95% CIs for the inter-observer agreement was quite wide with low bounds especially for the outcome measures of durations (Table 5). All p-values were below 0.05.

## 3.2 Approach and avoidance behaviour

Using dummy variables with sum contrasts allows us to interpret the main fixed effects stimulus and age even where the interaction is significant, and we can thus describe the most striking pattern for the different stimuli based on Figs 4 and 5.

 **3.2.1 Effect of stimulus on approach and avoidance behaviour.** Stimulus had a statistically highly significant effect on all outcome measures (Table 6). When confronted with the presumed positive stimuli Silage and Peat, pigs from both age groups spent less time in the Start Room and Avoidance Room and thus longer in the Runway where the stimuli were presented compared to all other stimuli. All animals but four (three piglets, one gilt) were in Contact with Silage and all but two (one piglet, one gilt) were in Contact with Peat. Both piglets and gilts spent more time in Contact with these two stimuli than with any other stimulus. Pigs showed fewest Compartment Changes and Escape Attempts when confronted with Peat, followed by the stimuli Silage and Mirror. Across all stimuli, pigs were in Contact with the presumed positive Mirror most often, but spent less time in contact with it compared to Peat and Silage. Pigs oriented towards the origin of the presumed positive Contact Calls more frequently and for longer durations than to other stimuli, but also oriented towards the other two auditory cues Farm Noises and Restriction Calls. All but one pig per age group were in Contact

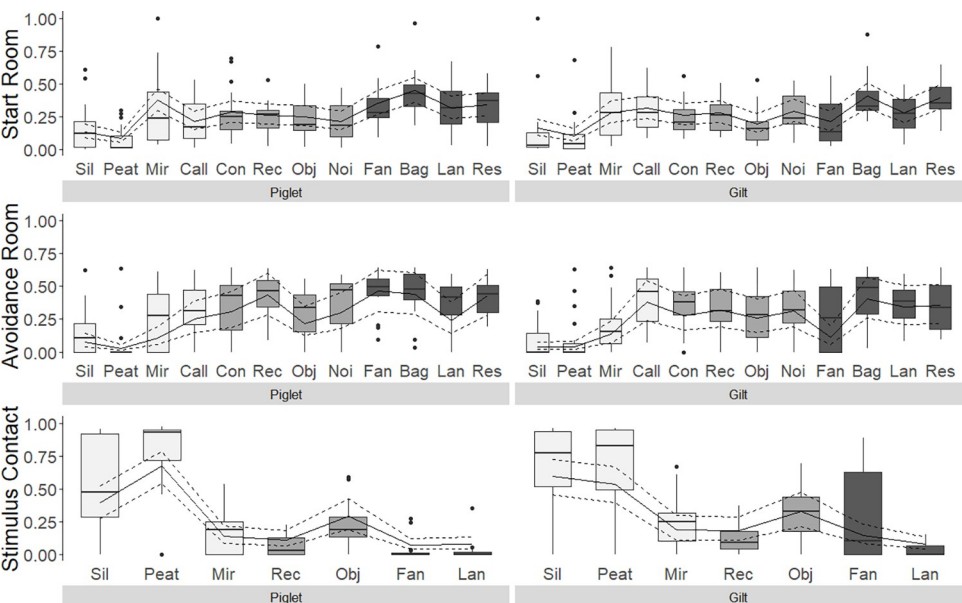

**Fig 4. Proportion of time spent in the Start Room or Avoidance Room or showing a behaviour (Stimulus Contact) presented across stimuli and separately for piglets and gilts.** Boxes show medians and the lower as well as upper interquartile range. Whiskers represent 1.5 times the interquartile range. Model estimates with estimated means (solid line) and 95% confidence intervals (dashed lines) are shown. White boxes: Presumed positive stimuli, light grey boxes: Presumed ambiguous stimuli, dark grey boxes: Presumed negative stimuli. Stimulus Contact: Contact Calls, Control, Farm Noises, Bag, Restriction Calls are not included. Sil: Silage, Mir: Mirror, Call: Contact Call, Con: Control, Rec: Rectangle, Obj: Object, Noi: Farm Noises, Lan: Lantern, Res: Restriction Calls.

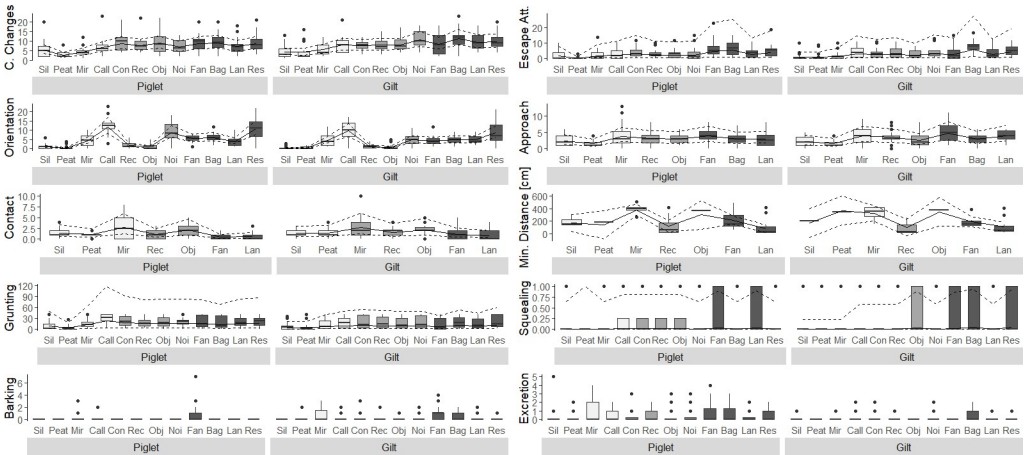

**Fig 5. Frequencies of different behaviours (Compartment Changes, Escape Attempts, Stimulus Orientation, Stimulus Approach, Stimulus Contact, Grunting, Barking, Excretion), the minimum distance to the stimulus if a pig had not been in contact with it and the proportion of animals that squealed presented across stimuli and separately for piglets and gilts.** Boxes show medians and the lower as well as upper interquartile range. Whiskers represent 1.5 times the interquartile range. Model estimates with estimated means (solid line) and 95% confidence intervals (dashed lines) are shown for all outcome measures where models converged (see Table 6). White boxes: Presumed positive stimuli, light grey boxes: Presumed ambiguous stimuli, dark grey boxes: Presumed negative stimuli. Orientation: Control not included. Approach: Contact Calls, Control, Farm Noises and Restriction Calls not included. Contact: Contact Calls, Control, Farm Noises, Bag, Restriction Calls not included. Minimal Distance: Contact Calls, Control, Farm Noises, Bag, Restriction Calls not included. Sil: Silage, Mir: Mirror, Call: Contact Call, Con: Control, Rec: Rectangle, Obj: Object, Noi: Farm Noises, Lan: Lantern, Res: Restriction Calls.

with the presumed ambiguous green plastic Object. The duration spent in Contact with the Object was similar as for the Mirror and longer compared to the remaining presumed ambiguous and negative stimuli. For the presumed negative stimuli, time spent in the Start Room and Avoidance Room (i.e. not the Runway) was longer compared to differently valanced stimuli, except for the Fan, where responses generally varied widely, especially between gilts. Ten out of 19 gilts were in Contact with the fan, but only five out of 20 piglets. Contact with the moving Bag was only short, but admittedly was also rather difficult for the pigs, which is why Bag was excluded from the analysis of the Stimulus Contact. At least one Escape Attempt was recorded in 72.0% of the sessions (336 of 467) and the number of Escape Attempts was highest for the presumably negative Fan (here only for piglets), the Bag and the Restriction Calls. Independent of age group, most elimination behaviour was recorded when pigs were confronted with the Bag (n = 12 sessions) or the Contact Calls (n = 10 sessions), whereas most Barking was displayed when pigs were exposed to the Mirror (n = 11 sessions) or Fan (n = 17), with the latter also inducing half of all startling responses (n = 9 sessions).

**3.2.2. Effect of age group on approach and avoidance behaviour.** After correcting for multiple testing, there was no statistically significant effect of age group on any outcome measure. Stimulus-specific differences between piglets and gilts are described below (3.2.3).

**3.2.3. Interacting effects of stimulus and age group on approach and avoidance behaviour.** The interaction between stimulus and age group had a statistically significant effect on five of the eleven analysed outcome measures (Table 6). The direction of these effects is derived from the graphical display of the data (Figs 4 and 5), which means that the described patterns are numerical and not necessarily statistically significant. Overall, the reactions of piglets and gilts towards the different stimuli was comparable, but with a few differences. Time spent in the Avoidance Room when exposed to the Mirror was seemingly higher for and varied more between piglets than gilts. Piglets also spent more time in the Avoidance Room when exposed

**Table 6. Model outcomes (p-value, test statistic) for the fixed effects stimulus, age group and their interaction.**

| Outcome measure | Stimulus*Age Group | | Stimulus | | Age Group | |
|---|---|---|---|---|---|---|
| | p-value[1] | test statistic | p-value[1] | test statistic | p-value[1] | test statistic |
| **Proportion of time** | | | | | | |
| Start Room | 0.05 | $D_{11} = 19.6$ | **< 0.001** | $D_{11} = 139.7$ | 0.87 | $D_1 = 0.03$ |
| Avoidance Room | **< 0.001** | $D_{11} = 51.7$ | **< 0.001** | $D_{11} = 341.7$ | 0.71 | $D_1 = 0.1$ |
| Stimulus Contact[2] | 0.23 | $D_{11} = 14.1$ | **< 0.001** | $D_{11} = 272.0$ | 0.14 | $D_1 = 2.2$ |
| **Frequencies** | | | | | | |
| Compartment Changes | **0.001** | $X^2_{11} = 33.3$ | **< 0.001** | $X^2_{11} = 277.3$ | 0.29 | $X^2_1 = 1.1$ |
| Escape Attempts | **< 0.001** | $X^2_{11} = 43.6$ | **< 0.001** | $X^2_{11} = 388.9$ | 0.94 | $X^2_1 = 0.006$ |
| Stimulus Orientation[3] | **0.004** | $X^2_{10} = 26.2$ | **< 0.001** | $X^2_{10} = 1171.95.8$ | 0.05 | $X^2_1 = 3.8$ |
| Stimulus Approach[4] | 0.80 | $X^2_7 = 3.8$ | **< 0.001** | $X^2_7 = 67.8$ | 0.64 | $X^2_1 = 0.2$ |
| Stimulus Contact[2] | 0.28 | $X^2_6 = 7.5$ | **< 0.001** | $X^2_6 = 87.8$ | 0.20 | $X^2_1 = 1.6$ |
| Grunting | **< 0.001** | $X^2_{11} = 85.8$ | **< 0.001** | $X^2_{11} = 687.7$ | 0.23 | $X^2_1 = 1.4$ |
| Squealing (0/1) | 0.16 | $X^2_{11} = 15.6$ | **< 0.001** | $X^2_{11} = 35.6$ | 1 | $X^2_1 = 0$ |
| Barking[5] (0/1) | | | | | | |
| Excretion[5] (0/1) | | | | | | |
| Startling[6] | | | | | | |
| **Distance to stimulus** | | | | | | |
| Min. Distance if no contact[7] | 0.71 | $F_{6,40} = 0.61$ | **< 0.0001** | $F_{6,40} = 10.6.16$ | 0.98 | $F_{1,1} = < 0.001$ |

[1] P-values smaller than 0.004 are considered statistically significant after Benjamini-Hochberg procedure

[2] Only stimuli for which physical Stimulus Contact was possible were included in this analysis, i.e. Silage, Peat, Mirror, Rectangle, Object, Fan, Lantern

[3] Control was excluded from this analysis since Stimulus Orientation could not be assessed when there was no stimulus.

[4] Only stimuli for which Stimulus Approach was possible were included in this analysis, i.e. Silage, Peat, Mirror, Rectangle, Object, Fan, Bag, Lantern

[5] Models did not converge, neither for frequency measures nor if data were treated as binomial.

[6] Pigs only startled in 18 times across all animals and test sessions, which is why this outcome measure was not further analysed

[7] Analysed for subset of stimuli for which physical contact was possible (see above) and only if animals did not have contact with the stimulus.

D: Deviance, $X^2$: Chi-square.

to the Fan, but here the variation between gilts was higher. Gilts seemed to show more Compartment Changes and to spend more time in Contact with the Fan than piglets, while piglets showed more Escape Attempts than gilts when confronted with this stimulus. When exposed to Contact Calls and Farm Noises, piglets oriented seemingly more often towards the loudspeakers than gilts. In addition, piglets grunted more than sows when hearing the Contact Calls and also in the Control situation. With respect to the behaviours indicative of anxiety or surprise for which statistical models could not be run, pigs barked in 11.6%, defecated/urinated in 18.0% and startled in 3.9% of the sessions. Even though we can only compare numerically, there were some apparent differences between both age group and stimuli. Whereas piglets defecated/urinated almost three times as often as gilts (61 sessions compared to 23), it was the other way around for Barking and Startling with gilts barking almost four times (43 compared to 11 sessions) and being startled in almost three times as many sessions as piglets (13 compared to 5 sessions).

## 4 Discussion

The aim of this study was to validate the valence of stimuli, which can then be used to differentiate between boredom and other negative states in pigs in future studies. Specifically, we aimed to assess how pigs respond to these stimuli without prior training or conditioning, a precondition for future tests, using an approach-avoidance paradigm. We found that pigs

showed different stimulus-specific behavioural responses regarding measures of approach and avoidance. For several stimuli the observed pattern was according to expectation, which means that pigs showed more frequent and longer behaviours associated with approach and less frequent and shorter behaviours associated with avoidance in presumed positive sessions and vice versa in presumed negative sessions, but this pattern was not consistent for all stimuli. The significant interactions between stimulus and age for some of the outcome measures indicates that age differences must be considered, but there was no consistent effect of age across all stimuli. Below we first discuss the results of the intra- and inter-observer agreement, then describe general considerations with respect to the outcome measures and finally interpret the identified patterns, including a discussion of individual differences. We conclude with an adapted classification of the valence of the stimuli.

## 4.1 Intra- and inter-observer agreement

Intra-observer agreement was excellent for most outcome measures with only a few exceptions, where it was good. Since only one observer (HH) coded all videos in this study, controlling for consistency in her coding was most important. Agreement between the two observers ranged from moderate to excellent for the outcome measures included in our analyses, with an overall better agreement for frequency than for duration measures. This indicates that behaviours could generally be well identified, but reaching agreement on start and end points was more difficult. Since at least some of the duration measures will also be relevant in future studies, for example the duration of Stimulus Contact, which was–together with the duration of Stimulus Orientation–identified as the most sensitive measure of stimulus interest in minks [8] and was also affected in ferrets [10], special care needs to be taken to clearly define start and end points. In our study, different potential end points were defined in the ethogram for some outcome measures, for example four different scenarios to end Stimulus Orientation, which may have increased discrepancies between observers.

## 4.2 Considerations for the interpretation of the approach and avoidance measures

Test paradigms aiming to differentiate between different negative states are based on the assumption that animals' affective states affect their interest in stimuli of differing valence. Thus, it would have been ideal to validate the stimuli in "neutral" animals housed in "neutral" conditions since otherwise the validation may be biased, for example if the study subjects are bored or very satisfied while being tested. We do not know the affective states of the animals tested in our study, but tried to find a compromise in the housing conditions by keeping both piglets and gilts in minimally enriched conditions with quite a high space allowance per pig and partially slatted floor. However, we cannot be sure how pigs' housing conditions affected our findings, which limits the generalisability of the results. If pigs in our study had been bored, we would expect an increased interest in all stimuli whereas if they had been apathetic, we would expect a decreased interest in all stimuli, independent of their valence. This means that the general level of interest would have been increased in case of boredom or decreased in case of apathy, but the general pattern and thus the substantial differences we found between stimuli of the different valences, would have remained. If pigs in our study had been depressed, we would have expected a decreased interest in positive stimuli due to anhedonia. However, we found strong differences between the presumably positive and other valenced stimuli.

More generally, it is possible that the farm management and other farm-specific characteristics affected how pigs responded to the stimuli. This limitation is not specific to our study, but refers to all studies conducted on one farm and could be counteracted by conducting multi-farm

experiments [33,34]. Besides potential farm-specific effects, the test situation itself probably affected pigs' affective states and thus potentially their reactions towards the stimuli. We tried to counteract this problem by keeping the test duration rather short (three minutes instead of originally planned five minutes) and by extensively habituating pigs first in groups and then in individual sessions until they stayed calm in the test apparatus (up to 17 habituation sessions).

Even though most outcome measures can be classified as either approach or avoidance behaviour, in order to gain insight into the bigger picture, it is important to interpret them together and not on their own since they depend on each other. A high frequency of Stimulus Contact, as for example seen when pigs were exposed to the Mirror, can be interpreted as positive since searching for contact clearly reflects interest in the stimulus. However, being only once or a few times in contact with a stimulus, but for a long time, as it was the case for Silage and Peat, is probably more indicative of the long-lasting positivity of the stimulus since it means that the pig has decided to stay in contact with it. Moreover, Stimulus Orientation is *per se* an indicator of approach or at least of attention, but is rare or short if the animal approaches the stimulus early in a test session and then stays in contact to it. In such a situation, a low level of Stimulus Orientation would even be interpreted as being positive. Another example demonstrating the importance of interpreting indicators together is the outcome measure Turning Away, which can only be shown if an animal has oriented to, approached or been in contact with the stimulus beforehand, which also explains the high correlation between Stimulus Approach and Turning Away. Thus, even though Turning Away on its own is an avoidance parameter, a high frequency of this behaviour indirectly reflects that the animal has previously shown some interest in the stimulus. Moreover,

Whereas some of the outcome measures may not be used in future studies, for example Turning Away due to its high correlation with Stimulus Approach, we suggest to add the latency to first contact as an additional outcome measure, which we did not record. Latency to first contact is a common measure used in Novel Object Tasks [e.g. 20] and has also been found to be longer in presumed negative compared to presumed positive stimuli in minks [9,13]. However, care needs to be taken since first, latency can only be applied to stimuli for which contact is possible and second, it is a censored measure (because there is no latency if pigs do not get in contact with the stimulus), which means that latency can only be analysed for sessions in which a pig got in contact with the stimulus.

With respect to the behavioural indicators of negative valence, we would exclude grunting in future studies since this kind of vocalisation does not allow clear interpretation of valence due to it being context-specific. However, a more detailed and potentially automatic analysis of different components of grunts and other calls may be promising with respect to valence [35].

Response patterns to the different stimuli and assessment of their valence

Aiming to interpret the different outcome measures together, it becomes apparent that the presumed positive Silage and Peat were indeed assessed as positive by the pigs since most pigs were in contact with these stimuli for the longest time across stimuli. A longer contact time with the positively classified stimuli compared to ambiguous and negative stimuli was also described by Meagher and Mason [9] and Meagher et al. [13] in mink. Moreover, Silage and Peat only elicited few avoidance behaviours like Compartment Changes or Escape Attempts. Silage and Peat both smell intensively, are edible and rootable, all important characteristics of stimulated motivation for exploration [36]. Moreover, added grains of corn possibly increased the positivity of the Peat, since it has been shown that such grains increase the attractiveness of straw in fattening pigs [21]. Reponses to the presumed positive Mirror showed a different pattern: Whereas pigs were frequently in contact with the Mirror, contact duration was rather short and the Minimal Distance of the pigs who did not have contact was large, indicating that variation between pigs was high. In addition, pigs showed some startle responses and barked

when exposed to the Mirror, with barking typically reflecting an alarming or surprising situation [27,37]. Taken together, behaviours shown when confronted with the Mirror ranged from approach, mostly characterised by interest as reflected by the high frequency of approaches, to avoidance measures and a high variability between pigs. These findings lead us to conclude that the Mirror cannot be used as a positive stimulus and that future research is needed to understand what exposure to a mirror means for pigs. The valence of the fourth presumed positive stimulus, the Contact Calls, is difficult to assess. This is probably due to the nature of sound as a stimulus, which also holds true for the other two auditory cues Farm Noises and Restriction Calls. They all elicited orientation towards the loud speakers, but orientation on its own is difficult to interpret in terms of valence, whereas it may be a good measure of attention. Moreover, some outcome measures like Stimulus Contact and Minimal Distance are difficult to use for the interpretation of the valence of auditory cues, because there is no object to be in contact with, which means that we have overall less information than for other stimuli. We still assessed how close pigs would come to the loud speakers, but since the stimulus, i.e. the sound, and the object where it originates from, i.e. the loudspeaker, are two different things, the validity of this measure is questionable when assessing auditory cues. Piglets grunted numerically more than gilts when exposed to the Contact Calls. These grunts were not necessarily a negative sign, but may have indicated that piglets paid more attention to this stimulus than gilts. This may be explained by the fact that the Contact Calls were grunts from piglets in a similar age as the piglets in our study, which may have stimulated more responding grunts in the piglets than in the older gilts. Since it is very difficult to interpret what the Contact Calls meant to the piglets who responded, but could not get access to the piglets that were the origin of the sound, we conclude that Contact Calls should not be used as a positive stimulus in future studies. Comparing all three auditory cues with each other, the pattern was quite similar across outcome measures, but the presumed negative Restriction Calls resulted in overall more avoidance behaviour.

Considering the presumed ambiguous stimuli, the patterns were overall as expected, that is in-between the presumed positive and negative stimuli, except for the green plastic Object, which differed from the other presumed ambiguous stimuli. All but one pig per age group were in contact with the Object and they spent longer time in direct contact with the Object compared to the other ambiguous stimuli, but also the Mirror. Pigs may have been more interested in the Object than in other presumed ambiguous stimuli since the Object was easier to manipulate, which may have maintained their interest. We thus conclude that the Object is perceived as rather positive than ambiguous.

Pigs spent more time in the Start Room and Avoidance Room (and thus less time in the Runway) when exposed to the presumed negative stimuli, especially the Bag filled with cans and the Restriction Calls. When confronted with these two stimuli, pigs also showed high numbers of Escape Attempts, leading us to conclude that they are interpreted as negative. Short time spent in the Runway, a high number of Escape Attempts and Barking as well as Startling were also recorded for the Fan, but interestingly more in piglets than in gilts. Fan is the stimulus where age differences became most apparent with gilts assessing the Fan as rather positive or at least interesting, whereas piglets perceived it as rather negative. Consequently, attention should be paid to the age of pigs in future tasks, especially when comparing results of pigs tested at a younger and again later at an older age.

### 4.3 Individual variation in responses

We expected pigs to vary more in their behavioural responses in presumed positive compared to negative trials, since the need for action should be higher and more comparable across

**Table 7. Classification of the presumed valence of the tested stimuli and the results of our study.**

| A | PRESUMED POSITIVE | | PRESUMED AMBIGUOUS | | PRESUMED NEGATIVE | |
|---|---|---|---|---|---|---|
| | Silage | | Control | | Fan | |
| | Peat | | Rectangle | | Bag | |
| | Mirror | | Object | | Lantern | |
| | Contact Calls | | Farm Noises | | Restriction Calls | |
| B | ASSESSED AS POSITIVE | | ASSESSED AS AMBIGUOUS | | ASSESSED AS NEGATIVE | |
| | Silage | | Control | | Fan piglets | Fan gilts |
| | Peat | | Rectangle | | Bag | |
| | Mirror | | Object | | Lantern | |
| | Contact Calls | | Farm Noises | | Restriction Calls | |

individuals in potentially dangerous negative situations, whereas individual differences should become more apparent in situations of opportunity, i.e. positive situations [38]. Additionally, variation should be highest in ambiguous situations, where uncertainty about the outcome is highest, as also seen in Judgement Bias Tasks where individual variation is higher when individuals are exposed to the uncertain ambiguous cues compared to the positive and negative reference cues, where the consequences have been learned and are thus certain [39]. However, this pattern was not found in our study, but variation rather depended on the specific stimuli and outcome measures than the presumed valence. High variation for some combinations of stimuli and outcome measures may result from differences in pigs' coping styles. Hessing and colleagues [40], for example, found that high reactive pigs showed more escape attempts, vocalised less and were less inhibited to approach, but explored less a novel object than low resisting pigs. Bolhuis and colleagues [41] also found differences in pigs' reactions to a novel object, depending on the interaction between coping style and housing treatment. Taking coping style into account may thus help explaining some of the variation between pigs in future studies.

## 5. Conclusions

Based on the results from our study, we suggest to adapt our classification as presented in Table 7.

Classification of the presumed valence of the tested stimuli (A) and the results of our study (B): Green indicates positive, yellow ambiguous and red negative valence. Light colours indicate that the evidence for a certain valence is less strong compared to bright colours. Grey means that it is difficult to allocate this stimulus to one valence. Stimuli for which the presumed valence was changed as a result of the study are indicated with bold font.

With the increasing interest in understanding the affective lives of farmed animals, future tests to differentiate between different negative states are warranted. We assessed how pigs react to different stimuli spontaneously, i.e. without prior learning, by applying an approach-avoidance paradigm, thereby paving the ground for future research aiming to differentiate between boredom, depression and apathic states in pigs.

## Supporting information

**S1 Fig. Proportion of time spent in the Start Room or Avoidance Room or showing a behaviour (Stimulus Contact) presented across stimuli and separately for female (n = 10) and male (n = 10) piglets.** Boxes show medians and the lower as well as upper interquartile range. Whiskers represent 1.5 times the interquartile range. White boxes: Presumed positive

stimuli, light grey boxes: Presumed ambiguous stimuli, dark grey boxes: Presumed negative stimuli.
(TIFF)

**S2 Fig. Frequencies of different behaviours (Compartment Changes, Escape Attempts, Stimulus Orientation, Stimulus Approach, Stimulus Contact, Grunting, Barking, Excretion), the minimum distance to the stimulus if a pig had not been in contact with it and the proportion of animals that squealed across stimuli and presented separately for female (n = 10) and male (n = 10) piglets.** Boxes show medians and the lower as well as upper interquartile range. Whiskers represent 1.5 times the interquartile range. White boxes: Presumed positive stimuli, light grey boxes: Presumed ambiguous stimuli, dark grey boxes: Presumed negative stimuli.
(TIFF)

**S1 File.**
(PDF)

## Acknowledgments

We would like to thank Daniela Kottik for installing the cameras, converting the videos and serving as second rater for the assessment of the inter-observer agreement. We are thankful to David Winckler for designing the apparatus and to him, Bruno Riha and Anna Oberpertinger for building it. We thankfully acknowledge Kurt Breu for assisting with running the test sessions and the Medau team for taking care of the animals. We would also like to thank Werner Mehlstäubl und Martina Knöbl for their advice with the programme Mangold INTERACT® and Sandra Düpjan and Lisette Leliveld for providing the different calls and noises used in this study.

## Author Contributions

**Conceptualization:** Sara Hintze, Heidi Heigl, Christoph Winckler.

**Formal analysis:** Sara Hintze.

**Funding acquisition:** Sara Hintze.

**Investigation:** Heidi Heigl.

**Writing – original draft:** Sara Hintze, Heidi Heigl.

**Writing – review & editing:** Sara Hintze, Heidi Heigl, Christoph Winckler.

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
