## [Decision Letter · Decision Letter 0]

3 Jul 2024

PONE-D-24-12879Towards a task to assess boredom-like states in pigs – stimulus validation as a basisPLOS ONE

Dear Dr. Hintze,

Thank you for submitting your manuscript to PLOS ONE. After careful consideration, we feel that it has merit but does not fully meet PLOS ONE’s publication criteria as it currently stands. Therefore, we invite you to submit a revised version of the manuscript that addresses the points raised during the review process. You will find the reviewer comments below.

We look forward to receiving your revised manuscript.

Kind regards,

I Anna S Olsson, Ph.D.

Academic Editor

PLOS ONE

Reviewers' comments:

Reviewer's Responses to Questions

**Comments to the Author**

1. Is the manuscript technically sound, and do the data support the conclusions?

Reviewer #1: Partly

Reviewer #2: Yes

2. Has the statistical analysis been performed appropriately and rigorously? 

Reviewer #1: I Don't Know

Reviewer #2: Yes

3. Have the authors made all data underlying the findings in their manuscript fully available?

Reviewer #1: Yes

Reviewer #2: Yes

4. Is the manuscript presented in an intelligible fashion and written in standard English?

Reviewer #1: Yes

Reviewer #2: Yes

5. Review Comments to the Author

Reviewer #1: The study is presenting a validation of the valence of twelve stimuli in pigs. The purpose of the study was to decide on an array of positive, negative and ambiguous stimuli that could be used in future studies of stimulus-perception in bored pigs compared to non-bored pigs. The study is very detailed and complex and the authors have without doubt done a thorough job.

I have some reservations and questions that I would like the authors to address.

General considerations

1.

Each stimulus was tested once (line 166). Only stimuli that the pig could intuitively decide the valence of without having to learn about “the consequences” of the stimuli was included (line 183).

The authors need to explain this. Some of the positive stimuli were food items such as silage and corn. Did the pigs have no prior experience with the scent of such food items? No experience from similar foods? The authors need to state in details what food- and enrichment items these pigs have been familiarized with during their early rearing. If the pigs can decide – on the basis of generalization from other known, smelling-foody stimuli – on the valence of the stimulus, then the stimulus is not really new-new. Prior experience and the ability of the pigs to generalize from that must be considered. That goes for all chosen stimuli.

The major flaw of this set-up is that previous experience and current housing is having an unknown effect on the mental state of the pigs. My experience is that pigs are different; some pigs will not eat apple slides at first; they need to go through several steps of apple-juice, apple-pieces-in-juice and small slides before realizing that apple slides are great… Other pigs just decide that apple slides are OK from the start. Also, pigs show an initial fright response to the mirror; but they will quickly start exploring… So there could also be issues of pig temperament and pig history influencing this behaviour. The authors do address this in the discussion, however, the resulting limitations on a future “boredom-assessment task” should be discussed.

The authors also use various calls and farm noises. How does that comply with the idea of using stimuli that the pigs have no prior experience with. Line 577-578: We assessed how pigs react to different stimuli they have not been in contact with before…”

2.

The authors need to discuss how it will influence the results if these gilts from an impoverished environment are already either bored or depressed.

3.

The pictures and illustrations need to be improved significantly. Please indicate openings in Figure 1. It is difficult (I think; perhaps I see them…) to see the yellow strips mentioned in Figure 2, etc.

Specific comments:

Introduction

Line 31. I think that If you decide that animals can be bored, then don’t fall into the pleaser-trap using the word “boredom-like”. Just write “…how animals experience boredom”.

Line 32: Suggest deleting “however”.

Line 43 and on: It would be nice to have a little more discussion on why the definition proposed by Meagher et al is the “chosen one”. There seem to be other definitions; please – in short - provide a few examples and justify the choice.

Line 72: Emotional, behavioural indicators (“signs of fear”) is mentioned for avoidance behaviours; please discuss if there are any useful indicators for approach behaviour (“signs of curiosity, play or whatever… just something …). Or do you consider an approach behaviour to inherently indicate a positive mental state (in opposition to neutral)?

Animals, material and methods

Line: 94: please add sex – so X 4-weeks old female piglets and y 4-weeks old, male piglets. This information may be mentioned somewhere else; I have not found it, though 

Line 100-108: Provide the bodyweight of the pigs (or mean bw in each group) and preferably a calculation of m2 per pig in each group. The pigs are housed in an impoverished environment; the space available is a housing factor tat should be included.

Line 142 and on: The decision criterias for the habituation phase is not clear. When was a pig considered ready to be on her own? Behaviural indicators? If a protocol for the habitation phase exist, consider adding it.

Line 287/section 2.8.3: Is it in any way possible to provide a table showing these decisions? It is a good job done, but difficult for the reader to understand.

Reviewer #2: In this manuscript, the authors aim to validate positive, neutral and negative stimuli to be used in behavioural tests for domestic pigs. This lays essential groundwork for future applied and fundamental research, opening the door to studies investigating the discrete states experienced by pigs. The manuscript is well written, and the rationale for the study is clearly outlined. I believe this will be a valuable contribution to the field of animal welfare science. However, there are a few areas where additional details are needed to ensure the replicability of this work, and I also have major concerns regarding the selection of outcome measures and potential influence of confounding/nuisance variables in this study. These are detailed below, followed by minor comments at the end.

Reporting: The authors have provided detailed descriptions of their methods, but given the complexity of the design (i.e., the number of groups and stimuli), there are a few areas where it would still be challenging to replicate this experiment. Please see minor comments below for specific details (e.g., regarding habituation and testing). However, as a broad recommendation, the authors are strongly encouraged to follow and cite the ARRIVE 2.0 Guidelines for reporting in this manuscript (as recommended by PlosOne). Attaching the ARRIVE Checklist in the supplementary material would be especially helpful.

Selection of Outcome Measures:

I have multiple concerns with the current approach taken for selecting outcome measures for the study. The first, and most worrying issue, is that the influence of stimulus valence on many of the indicators selected does not appear to be well understood. This is reflected in the fact that a large portion of the discussion highlights alternate interpretations of these indicators. In order to validate a stimulus as having a positive negative or neutral valance, it is essential that the relationship between valence and the outcome measures is well characterized a priori, so that logical predictions can be made from the hypothesis. Uncertainty in both the stimulus investigated and the outcome measures assessed risks circular reasoning.

The second issue is that there are numerous outcome measures assessed, but no primary outcome specified (see the ARRIVE Guidelones Essential 10, 6b). This makes it challenging for the reader to know how the outcomes should be “weighed” a priori (to avoid harking). Here, because the inferences about the valance of the stimulus are based on relative comparisons , ideally the primary outcome would be a measure that could be consistently applied across all stimuli (i.e., it would only be valance that drives differences in the outcome, not whether it was an auditory, visual etc. cue). However, if none of the outcome measures assessed meet this criteria, addressing gaps/weaknesses could be used to justify the secondary outcomes selected. Yet in line with this, it also seems that there should be careful consideration of which stimuli can be directly compared to each other, when stimuli type (e.g., auditory cues, manipulable objects) inherently trigger different responses from the animals (e.g., prolonged contact).

It also seems as though the many outcomes included could increase the risk of false negatives, since based on the information provided many of the outcomes may not reliably reflect approach/avoidance as intended. The authors have addressed the risk of false positives through a correction for multiple testing, but this risk of false negatives still seems concerning, particularly because of the lack of collinearity between the outcomes assessed (i.e., only a few outcomes described). If these outcomes are intended to measure the same things (approach and avoidance), are there expected reasons they do not covary (e.g., driven by different characteristics of the stimuli), or is it possible that some of the indicators selected could have been excluded based on poor sensitivity/specificity?

Identifying a primary outcome measure, justifying all secondary outcomes included, and carefully considering which stimuli can/should be directly compared to each other using each outcome measure is critical.

Controlling for Confounding/Nuisance Variables:

The following variables may not be adequately controlled for in the experimental design/statistical analyses. In many cases, additional details or adjustments to the analyses can address the concern. In instances where this is not the case, futher discussion of limitations may be needed.

The state of the pigs

Because the both piglets and gilts were housed in in minimally enriched conditions (ln 470-472), it seems the animals in the study will be at risk of negative welfare states identified in the introduction as influencing responses to stimuli (boredom, depression, apathy). Why were well-resourced, “enriched” animals not used for this study to reduce the risk of these states? And how might the generalizability of these findings be impacted if the pigs’ responses to stimuli were impacted by negative welfare states? This should be discussed/justified.

Neophobia

The authors did describe an extensive habituation phase in this study. However, it isn’t clear whether any data were collected during the habituation phase to confirm that the pigs had habituated to the apparatus (e.g., an asymptote in latency to leave the start area). Since the authors themselves highlight the importance of habituating animals in this study (ln 472-476), it should be clear whether there was evidence of habituation, or whether this was just assumed.

Random effects

The following random effects do not appear to be considered/assessed in the statistical analyses: Time of day, litter, group (I.e., based on the pen the animals were housed in). These should be considered, and use of a model selection process should be applied to justify which variables are excluded (e.g., excluding those that do not have significant effects on outcome variables, using forward or backward selection and assessing the impact on model fit – whichever approach the authors deem appropriate).

Time of day effects – pigs were tested between 6 and 23:00. Activity and appetite levels would naturally fluctuated within this range. Was this controlled for (e.g., by counterbalancing presentation of positive, negative and neutral stimuli across morning, afternoon, and night blocks) or investigated (e.g., checking for differences in responses to positive stimuli across time periods)? This also seems particularly important since two of your most positive stimuli (silage and peat) are edible.

Litter and housing group – Litter and housing group (i.e., the pen which animals were kept in) should be considered (using appropriate nesting with age, batch etc.) since animals from the same litter or home pen should not be treated as statistically independent (see Lazic 2010 https://doi.org/10.1186/1471-2202-11-5)

Sex effects – Just because they don’t have a hypothesis about sex effects, it does not mean it should not be controlled for/investigated. And at present, the graphical data shown in the supplementary material do not clearly demonstrate the absence of sex effects. In fact, they are presented in a way that makes it difficult for the reader to compare sexes within each stimuli (since they end up so far apart on the x-axis). At the very least, these should be plotted with male and female boxes beside each other for each stimulus, but statistical investigation would be more convincing.

Minor Comments:

Ln 46-48: It might be worth mentioning that that boredom, apathy and depression have been investigated together when animals demonstrate low arousal/activity responses to impoverished environments.

Ln 61: Is there a word missing after “was”? If not, the current phrasing is unclear.

Ln 82: Should “studies” be replaced with “stimuli”? Or does this refer to parallel experiments/studies?

Table 1: (and ln 93-98) Indicating the number of animals per crossbreed in Table 1 would be helpful to the reader to understand the experimental design.

Ln 142-143: It would be helpful to refer to the next section on habituation to be clear to the reader there will be more details provided.

Ln 149-162: With the level of detail provided, it would be difficult to replicate this habituation phase. Perhaps extending Table 2 to indicate the duration and group size combinations would help. It would also be beneficial to specify criteria for moving to a smaller group number or shorter habituation duration (“depending on pigs’ behaviour in the test apparatus” is quite vague). It seems this phase could have differed between groups and individuals, so was there any data collected to ensure animals were similarly habituated to the apparatus before testing began?

Ln 167-168: Specify method of randomization (e.g., random number generator, excel). Also, did each pig have a different random order? Or was the order randomized and presented this way for all pigs?

Ln 170-172: How were these sequences ensured to be different? Were they randomized? Was another approach for choosing the sequence taken?

Ln 191: “different” might be misleading here (suggesting the same stimuli could be played at different volumes for different pigs). Rephrasing for clarity could be helpful.

Ln 247-252: Were the observers (both live and video scoring) aware of the stimulus being presented and the presumed valence (i.e., was there any blinding to stimulus or hypothesis?)

Ln 303-305: I am not familiar with the approach of using dummy variables with sum contrasts to interpret main effects when interactions are significant, so cannot comment on this approach. But including a reference to justify/provide additional information for this method would be valuable.

Ln 384-385: Wording could be clearer here if “frequency” and “duration” were used as they were above.

Ln 446-448: If there was a significant interaction with age, doesn’t this suggest that age does significantly affect interpretation of stimuli?

Ln 452-465: It would be helpful to distinguish between agreement for live observations and video scoring throughout this section. I realize that intra-observer refers only to video scoring, and inter-observer refers to live scoring, but this is a bit tricky follow when reading the paragraph for the first time.

Ln 494-495: Were the tests video recorded? Or were videos only scored from a live feed to the monitor? If they were recorded, latency could potentially be scored and assessed (though only if the authors feel this would add clarity/be a viable primary outcome- see major comments above).

Ln 515: Suggesting use of the mirror as an ambiguous stimulus when the response from pigs was highly variable and difficult to interpret seems counterproductive. Calling for further investigation seems more valuable to future researchers than using it before it’s valence is understood.

Ln 525-527: Ef grunting frequency indicates “interest”, does this help with interpretation of valence? Or is this indicator similarly prone to reflecting attention rather than perceived valence?

6. PLOS authors have the option to publish the peer review history of their article (what does this mean?). If published, this will include your full peer review and any attached files.

Reviewer #1: No

Reviewer #2: No

---

## [Author Response · Author response to Decision Letter 0]

8 Aug 2024

We also uploaded a PDF document with all our responses, which we formatted for better readability. 

Response letter

to the two reviewers who reviewed the manuscript “Towards a task to assess boredom-like states in pigs – stimulus validation as a basis” submitted to PLOS ONE by Sara Hintze, Heidi Heigl and Christoph Winckler

Reviewer's Responses to Questions

Comments to the Author 1. Is the manuscript technically sound, and do the data support the conclusions? The manuscript must describe a technically sound piece of scientific research with data that supports the conclusions. Experiments must have been conducted rigorously, with appropriate controls, replication, and sample sizes. The conclusions must be drawn appropriately based on the data presented.

Reviewer #1: Partly

Reviewer #2: Yes

2. Has the statistical analysis been performed appropriately and rigorously?

Reviewer #1: I Don't Know

Reviewer #2: Yes

3. Have the authors made all data underlying the findings in their manuscript fully available? The PLOS Data policy requires authors to make all data underlying the findings described in their manuscript fully available without restriction, with rare exception (please refer to the Data Availability Statement in the manuscript PDF file). The data should be provided as part of the manuscript or its supporting information, or deposited to a public repository. For example, in addition to summary statistics, the data points behind means, medians and variance measures should be available. If there are restrictions on publicly sharing data—e.g. participant privacy or use of data from a third party—those must be specified.

Reviewer #1: Yes

Reviewer #2: Yes

4. Is the manuscript presented in an intelligible fashion and written in standard English? PLOS ONE does not copyedit accepted manuscripts, so the language in submitted articles must be clear, correct, and unambiguous. Any typographical or grammatical errors should be corrected at revision, so please note any specific errors here.

Reviewer #1: Yes

Reviewer #2: Yes

5. Review Comments to the Author Please use the space provided to explain your answers to the questions above. You may also include additional comments for the author, including concerns about dual publication, research ethics, or publication ethics. (Please upload your review as an attachment if it exceeds 20,000 characters)

We thank the reviewers for their constructive feedback. We have addressed each of the reviewers’ points individually and made notes indicating any changes to the manuscript. We believe doing so has substantially improved the manuscript and we hope it is now to your satisfaction.

Reviewer #1: The study is presenting a validation of the valence of twelve stimuli in pigs. The purpose of the study was to decide on an array of positive, negative and ambiguous stimuli that could be used in future studies of stimulus-perception in bored pigs compared to non-bored pigs. The study is very detailed and complex and the authors have without doubt done a thorough job. I have some reservations and questions that I would like the authors to address. General considerations 1. Each stimulus was tested once (line 166). Only stimuli that the pig could intuitively decide the valence of without having to learn about “the consequences” of the stimuli was included (line 183). The authors need to explain this. Some of the positive stimuli were food items such as silage and corn. Did the pigs have no prior experience with the scent of such food items? No experience from similar foods? The authors need to state in details what food- and enrichment items these pigs have been familiarized with during their early rearing. If the pigs can decide – on the basis of generalization from other known, smelling-foody stimuli – on the valence of the stimulus, then the stimulus is not really new-new. Prior experience and the ability of the pigs to generalize from that must be considered. That goes for all chosen stimuli.

Thank you for this remark. We aimed to only include stimuli for which the valence could be assessed on an ad hoc basis, i.e. spontaneously, without the need of prior learning. Pigs were not familiar with these types of feed as they were only used to standard feed. No pig was fed wet feed and there was no silage on the farm, so pigs were also not used to the smell; we received the silage from a collaborating dairy cow farm. Thus, we do not see a risk that pigs could generalise from their prior experience. This also applies to most of the other stimuli: No pig had prior experience with a mirror, a green plastic bowl/ball, a green wooden rectangle, a lantern, a bag with tins or a fan. However, pigs were used to farm noises and very likely to grunts and squeals. We agree that these stimuli were not new to them (as you criticise below). We don’t see this as a problem since these stimuli most likely had a comparable meaning for the pigs, without the need of learning. However, it is true though that prior experience to grunts/squeals may have influenced pigs’ assessment of these stimuli. Taking your consideration seriously, we have clarified that the important point is that pigs a) do not need to learn what a stimulus means first (revised sentence: “The most important selection criterion was that pigs could spontaneously assess the valence of the stimuli without the need to prior learn the meaning of a stimulus, as it is for example the case when they first need to get used to the taste of apples or chocolate”) and b) that it is important to avoid that pigs have different prior experience with the stimuli. We explicitly state now that the auditory cues were probably known to the pigs and that pigs’ experience with grunts and squeals may have slightly differed, but can be most likely be generalised across pigs (also from different farms) due to the inherent significance of these vocalisations: “All sounds from the three auditory stimuli were likely to be known by the pigs since they had probably encountered them earlier in their lives. Even though prior experience to these stimuli may have differed between pigs, grunts and squeals most likely had similar meaning to them due to the inherent significance of these calls for pigs.”

The major flaw of this set-up is that previous experience and current housing is having an unknown effect on the mental state of the pigs. My experience is that pigs are different; some pigs will not eat apple slides at first; they need to go through several steps of apple-juice, apple-pieces-in-juice and small slides before realizing that apple slides are great… Other pigs just decide that apple slides are OK from the start. Also,

pigs show an initial fright response to the mirror; but they will quickly start exploring… So there could also be issues of pig temperament and pig history influencing this behaviour. The authors do address this in the discussion, however, the resulting limitations on a future “boredom-assessment task” should be discussed.

We agree that pigs are different and also share your experience with getting them used to eat apples. This is the reason why it was important for us to use stimuli where this kind of learning that apples or chocolate taste good was not necessary. And indeed, silage and peat were assessed as positive, i.e. approached and consumed by almost all pigs without prior experience. As stated above, pigs were not experienced with these stimuli (with exception of the auditory cues, see above) and we thus do not see a risk for generalisation. Of course, pigs will always differ in the individual experience they have made before being tested in a task and individual differences will always be present. However, we counteracted the potential risk of individual differences biasing the results by including quite a large number of pigs for a within-subject design.

We agree that we do not know how housing conditions affected the pigs. However, we tried our best in finding a “middle way” of keeping pigs minimally enriched in order to avoid too strong effects coming from the housing conditions. We are not aware what we could have done in order to secure a neutral state of the pigs. However, we added the limited generalisability and an explanation how potential boredom, apathy or depression would have affected our results in the Discussion, as requested by you below.

One way of receiving more generalisable results are multi-farm studies, which help to exclude any influences coming from the management, the micro climate, etc. However, this advantage holds true for all studies and is not specific for ours. We added this suggestion in the Discussion.

The revised Discussion now reads: “However, we cannot be sure how pigs’ housing conditions affected our findings, which limits the generalisability of the results. If pigs in our study had been bored, we would expect an increased interest in all stimuli whereas if they had been apathetic, we would expect a decreased interest in all stimuli, independent of their valence. This means that the general level of interest would have been increased in case of boredom or decreased in case of apathy, but the general pattern, i.e. the substantial differences we found between stimuli of the different valences, would have remained. If pigs in our study had been depressed, we would have expected a decreased interest in positive stimuli due to anhedonia. However, we found strong differences between the presumably positive and other valenced stimuli.

More generally, it is possible that the farm management and other farm-specific characteristics affected how pigs responded to the stimuli. This limitation is not specific to our study, but refers to all studies conducted on one farm and could be counteracted by conducting multi-farm experiments (Nawroth & Gygax, 2021; Voelkl et al., 2018).”

The authors also use various calls and farm noises. How does that comply with the idea of using stimuli that the pigs have no prior experience with. Line 577-578: We assessed how pigs react to different stimuli they have not been in contact with before…”

Please see our comment above; we now describe in the Methods that pigs had most likely prior experience with components of the auditory cues, that their experience may have been different, but that grunts and squeals are of inherent significance for pigs.

2. The authors need to discuss how it will influence the results if these gilts from an impoverished environment are already either bored or depressed.

Please see our comment and the changes we made in the manuscript above.

3. The pictures and illustrations need to be improved significantly. Please indicate openings in Figure 1. It is difficult (I think; perhaps I see them…) to see the yellow strips mentioned in Figure 2, etc.

Yes, we agree that the quality is very low in the created PDF; however, it is good when you open the figures with the link on the top right corner of each figure (“Click here…”).

The openings are indicated in Figure 1, but they were not mentioned in the legend; we added this information now: “Fig 1. Schematic drawing of the test apparatus from the side (A) and top (B). Positions of the cameras (red square and arrow) and microphones (blue dots) are shown. Black arrows with capital letters indicate the camera perspectives from which the pictures presented in Fig 2A-E were taken. The guillotine doors between the Start Room and the Runway (also shown in Fig 2A and 2D (closed) and 2G (opened)), between the Start Room and the Avoidance Room (also shown in Fig 2A and 2B (closed) and 2G (opened)) and at the end of the Runway where the stimuli were presented (also shown in Fig 2C and 2D (closed) and 2E (opened)) are indicated as [-]. Dashed lines indicate the ropes used to open the guillotine doors via a pulley system (also visible in Fig 2D).”

The yellow stripes in Figure 2 are well visible in the original figure (see above).

Specific comments: Introduction Line 31. I think that If you decide that animals can be bored, then don’t fall into the pleaser-trap using the word “boredom-like”. Just write “…how animals experience boredom”.

Thank you for raising this point, which we understand. However, we did not use the term “boredom-like states” to be careful if or if not pigs can experience boredom, but because we cannot be sure that human boredom is identical to animal boredom. This is why we would like to stick to “boredom-like states”.

Line 32: Suggest deleting “however”.

We are not sure why you propose deleting “however” in this line? In the first sentence we state that “there is a lack of research” and in the second we show the contrast of many animals being kept in barren and monotonous conditions. We thus think that “however” fits well here to underline this contrast?

Line 43 and on: It would be nice to have a little more discussion on why the definition proposed by Meagher et al is the “chosen one”. There seem to be other definitions; please – in short - provide a few examples and justify the choice.

The few conceptual publications on animal boredom are all derived from human psychology. To our knowledge, no definition of animal boredom that is independent of human research exists. We chose the definition proposed by Meagher and colleagues since this is an operational definition aiming to empirically study boredom in non-human animals, forming the basis for the development of a task aiming to differentiate between boredom, depression and apathy and thus being the basis of our experiment. To clarify this, we have now added the most relevant elements of the definition of human boredom in the Introduction: “Definitions of animal boredom have been derived from human psychology. For example, boredom in humans has been characterised by disengagement, high arousal, low arousal, inattention and altered time perception (Fahlman et al., 2013), five characteristics that may play a role in animal boredom, too (Burn, 2017). Moreover, boredom has been described as a lack of meaningful engagement (Danckert & Eastwood, 2020). Based on these definitions of human boredom and aiming to study boredom In their empirically in mink studies, Meagher and colleagues operationally defined boredom as a negative state caused by a barren environment that results in an increased interest in all kinds of stimuli, independent of their valence, i.e. independent of whether the stimuli are perceived as positive, ambiguous or negative (Meagher et al., 2017; Meagher & Mason, 2012).”

Line 72: Emotional, behavioural indicators (“signs of fear”) is mentioned for avoidance behaviours; please discuss if there are any useful indicators for approach behaviour (“signs of curiosity, play or whatever… just something …). Or do you consider an approach behaviour to inherently indicate a positive mental state (in opposition to neutral)?

Thank you for raising this point. We now differentiate between approach/avoidance measures and behavioural indicators that provide additional information on pigs’ affective states, i.e. in case of our outcome measures on the negativity of the situation. We clarified this differentiation in the Methods:

“In addition to classical measures of approach and avoidance, we also included behavioural measures indicative of negative affective states (different vocalisations, elimination behaviour and startling) aiming to gain additional insight in what exposure to the different stimuli means to the pigs. We could not include behavioural measures indicative of positive affect due to a lack of validated positive indicators.”

Indeed, it would have been very nice to include also positive behavioural indicators, but despite all the research efforts currently done on positive animal welfare, we do not yet have validated behavioural indicators, whereas the negative measures we selected have been validated in different circumstances. Of course, it would be great to record curiosity, but in order to avoid circular reasoning, we needed to choose validated indicators and we are not aware of an indicator for curiosity, besides showing approach behaviour.

Animals, material and methods Line: 94: please add sex – so X 4-weeks old female piglets and y 4-weeks old, male piglets. This information may be mentioned somewhere else; I have not found it, though

Thank you for spotting this. The in

---

## [Decision Letter · Decision Letter 1]

13 Sep 2024

PONE-D-24-12879R1Towards a task to assess boredom-like states in pigs – stimulus validation as a basisPLOS ONE

Dear Dr. Hintze,

Thank you for submitting your manuscript to PLOS ONE. After careful consideration, we feel that it has merit but does not fully meet PLOS ONE’s publication criteria as it currently stands. Therefore, we invite you to submit a revised version of the manuscript that addresses the points raised during the review process. The remaining minor issues are outlined at the end of this e-mail.

We look forward to receiving your revised manuscript.

Kind regards,

I Anna S Olsson, Ph.D.

Academic Editor

PLOS ONE

Journal Requirements:

Reviewers' comments:

Reviewer's Responses to Questions

**Comments to the Author**

1. If the authors have adequately addressed your comments raised in a previous round of review and you feel that this manuscript is now acceptable for publication, you may indicate that here to bypass the “Comments to the Author” section, enter your conflict of interest statement in the “Confidential to Editor” section, and submit your "Accept" recommendation.

Reviewer #2: (No Response)

2. Is the manuscript technically sound, and do the data support the conclusions?

Reviewer #2: Yes

3. Has the statistical analysis been performed appropriately and rigorously? 

Reviewer #2: Yes

4. Have the authors made all data underlying the findings in their manuscript fully available?

Reviewer #2: Yes

5. Is the manuscript presented in an intelligible fashion and written in standard English?

Reviewer #2: Yes

6. Review Comments to the Author

Reviewer #2: Thank you to the authors for their thorough responses to my previous comments, as well as for the corresponding revisions made in the manuscript. The additional details and rationale provided have greatly enhanced the clarity of this study and addressed my primary concerns.

In particular, I appreciate the improvements made to the completeness of reporting, which effectively demonstrate the value of this work. The use of the ARRIVE Guidelines and the inclusion of the ARRIVE Essential 10 Checklist were valuable additions and the updates to the text, figures, and methods ensure that the study can be better understood and replicated by future researchers.The additional information regarding the criteria for confirming pigs were habituated, as well as the updated plots on sex differences, made these aspects of the study much easier to interpret. These updates also offered reassurance that there were no significant differences between groups, which strengthens the findings.

The rationale for the selection of stimuli and outcomes is now clearer. While it remains somewhat surprising that the chosen stimuli and outcomes do not consistently facilitate comparison, this is now transparently communicated in a way that allows readers to better determine which stimuli might be suitable for comparison in future studies.

I have only a few minor comments for this revised version of the manuscript:

• Line 528: Would it be possible to provide a value showing the space per pig in this study and compare it to other studies or requirements? For instance, providing evidence that these conditions were more likely to induce a negative state would strengthen the discussion.

• Adding a legend for the colors used in Figures 4 and 5 (box plots) would be valuable for clarity.

• Line 248-249: It might be easier to interpret if you indicate that the control is the empty runway at the first mention. For example, “encompassed a Control (the empty runway), a green….”, or something similar.

7. PLOS authors have the option to publish the peer review history of their article (what does this mean?). If published, this will include your full peer review and any attached files.

Reviewer #2: No

---

## [Author Response · Author response to Decision Letter 1]

15 Sep 2024

Response letter 2 

to the reviewer who re-reviewed the manuscript “Towards a task to assess boredom-like states in pigs – stimulus validation as a basis” submitted to PLOS ONE by Sara Hintze, Heidi Heigl and Christoph Winckler

Reviewer #2: Thank you to the authors for their thorough responses to my previous comments, as well as for the corresponding revisions made in the manuscript. The additional details and rationale provided have greatly enhanced the clarity of this study and addressed my primary concerns.

In particular, I appreciate the improvements made to the completeness of reporting, which effectively demonstrate the value of this work. The use of the ARRIVE Guidelines and the inclusion of the ARRIVE Essential 10 Checklist were valuable additions and the updates to the text, figures, and methods ensure that the study can be better understood and replicated by future researchers. The additional information regarding the criteria for confirming pigs were habituated, as well as the updated plots on sex differences, made these aspects of the study much easier to interpret. These updates also offered reassurance that there were no significant differences between groups, which strengthens the findings.

The rationale for the selection of stimuli and outcomes is now clearer. While it remains somewhat surprising that the chosen stimuli and outcomes do not consistently facilitate comparison, this is now transparently communicated in a way that allows readers to better determine which stimuli might be suitable for comparison in future studies.

Thank you for your positive feedback and, especially, for highlighting which aspects of our revision you appreciate and why. Below we respond to your additional comments in blue.

I have only a few minor comments for this revised version of the manuscript:

• Line 528: Would it be possible to provide a value showing the space per pig in this study and compare it to other studies or requirements? For instance, providing evidence that these conditions were more likely to induce a negative state would strengthen the discussion.

Thank for you this thought. We are a bit surprised about this line of reasoning since you asked us in the first round of reviewing why animals were not kept in fully enriched conditions, but would like us now to strengthen the argumentation by stating that pigs were in a rather negative state (which was criticised before). We would still argue that we aimed to find a baseline by keeping pigs in as “neutral” conditions as somehow possible, thereby avoiding both “too good” and “too bad” conditions. With respect to the available space allowance, we have argued that pigs in our study had more space per pig than under conventional conditions, and that this higher space allowance in combination with a little amount of sawdust (piglets) and straw (gilts) was what we defined as “minimally enriched”.

Besides this, we could compare space allowances across studies, but we know that space allowance is only one of several factors affecting pigs’ affective states with especially enrichment being of paramount importance for this curios species that spends three quarters of their active time exploring, rooting and feeding. Thus, even though we would be very happy if we could provide more information on the pigs’ affective welfare by putting the conditions of our pigs into perspective, we don’t think that comparing space allowances will help with it.

• Adding a legend for the colors used in Figures 4 and 5 (box plots) would be valuable for clarity.

We agree with you and complemented the legends of Figures 4 and 5 accordingly: “White boxes: presumed positive stimuli, light grey boxes: presumed ambiguous stimuli, dark grey boxes: presumed negative stimuli.” We also added this information for the figures in the Supplementary Information.

• Line 248-249: It might be easier to interpret if you indicate that the control is the empty runway at the first mention. For example, “encompassed a Control (the empty runway), a green….”, or something similar

Thank you for this suggestion. We adapted the sentence, which now reads: “The four stimuli defined as potentially ambiguous (Fig 3F-K) encompassed a Control, i.e. the empty Runway, a green Rectangle, a green plastic Object, and an audio recording of Farm Noises.”

---

## [Editor Report · Decision Letter 2]

25 Sep 2024

Towards a task to assess boredom-like states in pigs – stimulus validation as a basis

PONE-D-24-12879R2

Dear Dr. Hintze,

We’re pleased to inform you that your manuscript has been judged scientifically suitable for publication and will be formally accepted for publication once it meets all outstanding technical requirements.

Kind regards,

I Anna S Olsson, Ph.D.

Academic Editor

PLOS ONE
---

## [Editor Report · Acceptance letter]

3 Oct 2024

PONE-D-24-12879R2 

PLOS ONE

Dear Dr. Hintze, 

I'm pleased to inform you that your manuscript has been deemed suitable for publication in PLOS ONE. Congratulations! Your manuscript is now being handed over to our production team.

Kind regards, 

on behalf of

Dr. I Anna S Olsson 

Academic Editor

PLOS ONE